# Maximum Mean Discrepancy with Unequal Sample Sizes via Generalized U-Statistics

**Aaron Wei**                                                                          *aaronwei221@gmail.com*
*University of British Columbia*

**Milad Jalali**                                                                        *jalalim807@gmail.com*
*Independent researcher, Vancouver, Canada*

**Danica J. Sutherland**                                                                 *dsuth@cs.ubc.ca*
*University of British Columbia & Alberta Machine Intelligence Institute*

**Reviewed on OpenReview:** *https://openreview.net/forum?id=KjXW75GHHF*

## Abstract

Existing two-sample testing techniques, particularly those based on choosing a kernel for the Maximum Mean Discrepancy (MMD), often assume equal sample sizes from the two distributions. Applying these methods in practice can require discarding valuable data, unnecessarily reducing test power. We address this long-standing limitation by extending the theory of generalized U-statistics and applying it to the usual MMD estimator, resulting in new characterization of the asymptotic distributions of the MMD estimator with unequal sample sizes (particularly outside the proportional regimes required by previous partial results). This generalization also provides a new criterion for optimizing the power of an MMD test with unequal sample sizes. Our approach preserves all available data, enhancing test accuracy and applicability in realistic settings. Along the way, we give much cleaner characterizations of the variance of MMD estimators, revealing something that might be surprising to those in the area: while zero MMD implies a degenerate estimator, it is sometimes possible to have a degenerate estimator with nonzero MMD as well. We give a construction of such a case, and a proof that it does not happen in common situations.

## 1 Introduction

Two-sample testing is a fundamental problem in statistical inference, where the objective is to determine whether two arbitrary distributions, $P$ and $Q$, differ, based only on samples drawn from each. Applications include distinguishing treatment and control groups (e.g. Kobayashi et al., 2017), evaluating and training generative models (e.g. Li et al., 2015; Dziugaite et al., 2015; Bińkowski et al., 2018; Jayasumana et al., 2024), and domain adaptation (e.g. Long et al., 2013), among many others. Given a dataset $\mathbf{S} := (\mathbf{S}_P, \mathbf{S}_Q)$, where $\mathbf{S}_P := (x_i)_{i \in [n_X]} \sim P^{n_X}$ and $\mathbf{S}_Q = (y_i)_{i \in [n_Y]} \sim Q^{n_Y}$, the goal is to test the null hypothesis $H_0 : P = Q$ versus the alternative $H_1 : P \neq Q$. Typically, we do this by comparing a test statistic $\tau(\mathbf{S})$ to a threshold $c_\alpha$ for a given significance level $\alpha$, say 0.05. The null hypothesis is rejected if $\tau(\mathbf{S}) > c_\alpha$. The threshold should be set such that if $H_0$ holds, the probability of $\tau(\mathbf{S})$ exceeding $c_\alpha$ (the false rejection rate) is at most $\alpha$.

In recent years, tests based on the kernel maximum mean discrepancy or MMD (Gretton, Borgwardt, et al., 2012), or equivalently (Sejdinovic et al., 2013) tests based on energy statistics (Székely and Rizzo, 2013), have gained popularity due to their flexibility, adaptivity to various settings, ease of implementation, computational simplicity, and desirable theoretical properties. Setting the threshold $c_\alpha$ is now usually addressed by permutation testing (Sutherland et al., 2017; Hemerik and Goeman, 2018) rather than the asymptotic distribution, for both computational and statistical reasons. Methods for choosing a kernel that will perform well on the particular task at hand, however, are mostly based on an estimate of the asymptotic

behavior of the test statistic under both the null and the alternative distributions (Gretton, Sejdinovic, et al., 2012; Sutherland et al., 2017; Liu et al., 2020; Kübler, Stimper, et al., 2022; Deka and Sutherland, 2023).

When the number of samples from both distributions is equal, an MMD estimator can be constructed as a U-statistic. This allows for the application of well-established asymptotic results for U-statistics (see e.g. Lee, 2019) in designing the testing procedure, as has been done in practice for the line of work cited above (excepting that of Kübler, Stimper, et al. (2022), who use a different and more limited framework).

In practice, however, sample sizes are often unequal, leading to an MMD estimator that is not a U-statistic. Consequently, the theoretical framework developed for the equal sample size case cannot be directly applied. (Even when sample sizes are equal, the U-statistic estimator is slightly different from the usual, lower-variance, unbiased estimator.) Liu et al. (2020) addressed this by training on equal-sized subsamples, finding a kernel which has roughly the best test power for a test between samples of equal size, then perhaps applying that kernel to a test set with differing sizes. While this procedure works, it is wasteful.

Other approaches, based on different theoretical foundations, do not have such issues: the most dominant category which allows for learning representations to handle difficult testing problems is that of classifier-based two sample tests (Lopez-Paz and Oquab, 2017; Kim et al., 2021; Kübler, Stimper, et al., 2022), but these are often not as strong tests as the more general MMD-based tests (Liu et al., 2020, Section 4). Theory of the related cross-MMD estimator (Shekhar et al., 2023) does not make the equal-sample-size assumption, but nor does it provide a method for selecting kernels.[1]

This paper addresses this limitation by deriving the asymptotic distributions and variance estimators for the MMD estimator under unequal sample sizes. As noted by Kim et al. (2022), the typical MMD estimator can be viewed as a *generalized* U-statistic, but the asymptotics of this more general framework are as yet underdeveloped. We find the required general results and apply them to the MMD estimator to find its asymptotic distribution under both null and alternative hypotheses, allowing us to choose kernel tests to maximize their power (Sutherland et al., 2017; Liu et al., 2020; Deka and Sutherland, 2023). Along the way, we also correct a common misconception in the literature about MMD under the alternative hypothesis.

Furthermore, we demonstrate that our generalized testing procedure reduces to the previous approach when sample sizes are equal, thus establishing it as a natural extension of the existing framework. This unification of methods provides a comprehensive approach to two-sample testing using MMD, accommodating a wider range of practical scenarios while maintaining theoretical rigor.

## 2 Preliminaries

The Maximum Mean Discrepancy (MMD) is a widely used metric to measure the distance between probability distributions. Its versatility and theoretical properties make it particularly useful in various statistical applications, including two-sample testing.

### 2.1 Kernel mean embeddings and Hilbert-Schmidt operators

Let $k : \mathcal{X} \times \mathcal{X} \to \mathbb{R}$ be the kernel of a reproducing kernel Hilbert space (RKHS) $\mathcal{H}$ of functions $\mathcal{X} \to \mathbb{R}$; the notation $k(\cdot, x)$ denotes the element $t \mapsto k(t, x)$, which satisfies the reproducing property $\langle f, k(\cdot, x) \rangle_{\mathcal{H}} = f(x)$ for all $f \in \mathcal{H}$, and hence $k(x, y) = \langle k(\cdot, x), k(\cdot, y) \rangle_{\mathcal{H}}$. The *mean embedding* of a distribution $P$ is

$$\mu_P = \mathbb{E}_{X \sim P}[k(X, \cdot)] \in \mathcal{H}.$$

The mean embedding exists and is well-defined when $\mathbb{E}\|k(X, \cdot)\| = \mathbb{E}\sqrt{k(X, X)} < \infty$; it satisfies a reproducing property for distributions, $\langle f, \mu_P \rangle_{\mathcal{H}} = \mathbb{E}_{X \sim P} f(X)$ for any $f \in \mathcal{H}$. A related concept is the (centered)

---

[1]The primary advantage of the cross-MMD, as opposed to the more traditional MMD estimator considered here, is computational: it avoids the need for permutation testing through a simpler null distribution. In regimes where we are conducting kernel learning, however, permutation testing is typically not a particularly dominant part of the computational cost: choosing the (possibly deep) kernel parameters is much more expensive. Thus, while the power of a cross-MMD test is closely related to the power of a "full" MMD test and hence our approach for kernel learning could be used to learn the kernel of a cross-MMD test, there does not seem to be much practical reason to do so.

*covariance operator* of a distribution $P$, given by

$$C_P = \mathop{\mathbb{E}}_{X \sim P}[k(X, \cdot) \otimes k(X, \cdot)] - \mu_P \otimes \mu_P,$$

which exists as long as $\mathbb{E}\, k(X, X) < \infty$. Here the outer product $f \otimes g$ for $f, g \in \mathcal{H}$ is viewed as an $\mathcal{H} \to \mathcal{H}$ operator with $[f \otimes g](g') = f\langle g, g'\rangle$, so that $\langle f, C_P g\rangle = \text{Cov}_{X \sim P}(f(X), g(X))$ for any $f, g \in \mathcal{H}$. The elements $f \otimes g$ and $C_P$ are Hilbert-Schmidt operators from $\mathcal{H} \to \mathcal{H}$, denoted by $f \otimes g \in \text{HS}(\mathcal{H}, \mathcal{H})$; this is itself a Hilbert space with inner product $\langle f \otimes f', g \otimes g'\rangle_{\text{HS}} = \langle f, g\rangle_{\mathcal{H}}\langle f', g'\rangle_{\mathcal{H}}$. The review of Muandet et al. (2017) describes these two objects further. Throughout this paper, we will assume that they are well-defined:

**Setting A.** *We assume that $k : \mathcal{X} \times \mathcal{X} \to \mathbb{R}$ is a kernel which induces a separable RKHS $\mathcal{H}$, and that the covariance operator $C_P$ and mean embedding $\mu_P$ exist for all considered distributions, i.e. $\mathbb{E}_{X \sim P}\, k(X, X) < \infty$.*

For separability, it suffices to have a separable $\mathcal{X}$ and continuous $k$ (Steinwart and Christmann, 2008, Lemma 4.33). This gives us a variety of results, such as the following.

**Proposition 2.1.** *In Setting A, for independent random variables $X \sim P, Y \sim Q$, we have $\mathbb{E}[k(X, Y)^2] < \infty$.*

*Proof.* Since $\mu_P, C_P$ exist, the uncentered covariance operator $\tilde{C}_P = C_P + \mu_P \otimes \mu_P = \mathbb{E}[k(\cdot, X) \otimes k(\cdot, X)]$ is Hilbert-Schmidt, and likewise for $\tilde{C}_Q = \mathbb{E}[k(\cdot, Y) \otimes k(\cdot, Y)]$. By Bochner integrability, we can move expectations in and out of inner products, obtaining that the following inner product must be finite:

$$\langle \tilde{C}_P, \tilde{C}_Q\rangle_{\text{HS}} = \left\langle \mathop{\mathbb{E}}_X k(\cdot, X) \otimes k(\cdot, X), \mathop{\mathbb{E}}_Y k(\cdot, Y) \otimes k(\cdot, Y)\right\rangle_{\text{HS}} = \mathbb{E}\big[\langle k(\cdot, X), k(\cdot, Y)\rangle_{\mathcal{H}}^2\big] = \mathbb{E}\big[k(X, Y)^2\big]. \qquad \square$$

## 2.2 Maximum Mean Discrepancy

**Definition 2.2.** *Given a reproducing kernel Hilbert space $\mathcal{H}$, the MMD between distributions $P$ and $Q$ is defined as*

$$\text{MMD}(P, Q) := \sup_{f \in \mathcal{H}: \|f\|_{\mathcal{H}} \le 1} \mathop{\mathbb{E}}_{X \sim P} f(X) - \mathop{\mathbb{E}}_{Y \sim Q} f(Y).$$

The MMD always satisfies all properties of a metric except that it may have $\text{MMD}(P, Q) = 0$ for some $P \ne Q$; this does not happen for *characteristic* kernels (Sriperumbudur et al., 2011). None of our results will require a characteristic kernel.

Using the reproducing property, under Setting A MMD can be rewritten as (Gretton, Borgwardt, et al., 2012)

$$\text{MMD}^2(P, Q) = \|\mu_P - \mu_Q\|_{\mathcal{H}}^2 = \mathop{\mathbb{E}}_{X, X' \sim P}[k(X, X')] + \mathop{\mathbb{E}}_{Y, Y' \sim P}[k(Y, Y')] - 2\mathop{\mathbb{E}}_{X \sim P, Y \sim Q}[k(X, Y)].$$

Estimators for the MMD are often based on this last form. Assuming independent samples $\mathbf{S}_P := \{x_i\}_{i \in [n_X]} \sim P^{n_X}$ and $\mathbf{S}_Q = \{y_i\}_{i \in [n_Y]} \sim Q^{n_Y}$, the typical unbiased estimator is

$$\widehat{\text{MMD}}^2 = \frac{2}{n_X(n_X - 1)} \sum_{i=1}^{n_X} \sum_{j=i+1}^{n_X} k(x_i, x_j) + \frac{2}{n_Y(n_Y - 1)} \sum_{i=1}^{n_Y} \sum_{j=i+1}^{n_Y} k(y_i, y_j) - \frac{2}{n_X n_Y} \sum_{i=1}^{n_X} \sum_{j=1}^{n_Y} k(x_i, y_j). \quad (1)$$

If $n_X = n_Y = n$, we can obtain a simpler, nearly-equivalent estimator. Let $z_i = (x_i, y_i)$, and define

$$h(z_i, z_j) = k(x_i, x_j) + k(y_i, y_j) - k(x_i, y_j) - k(x_j, y_i) \qquad (2)$$

$$\widehat{\text{MMD}}_{\text{U}}^2 = \frac{1}{n(n-1)} \sum_{i \ne j} h(z_i, z_j). \qquad (3)$$

This estimator differs from $\widehat{\text{MMD}}^2(\mathbf{S}_P, \mathbf{S}_Q)$ only in that it omits the $k(x_i, y_i)$ terms; this remains unbiased for $\text{MMD}^2$, but has very slightly higher variance (compare Proposition 2.6 and Theorem 3.4) and unlike $\widehat{\text{MMD}}^2$ depends on the order of the two samples. The difference can be directly bounded by McDiarmid's inequality (details in Appendix A).

**Theorem 2.3.** *When the number of samples from $P$ and $Q$ are the same, we have that*

$$\Pr_{\substack{X_1,\ldots,X_n \sim P \\ Y_1,\ldots,Y_n \sim Q}} \left( |\widehat{\mathrm{MMD}}^2 - \widehat{\mathrm{MMD}}_\mathrm{U}^2| \leq \frac{8 \sup_{x \in \mathcal{X}} k(x,x)}{n^{3/2}} \sqrt{\log \frac{2}{\delta}} \right) \geq 1 - \delta.$$

### 2.3 U-Statistics

The estimator $\widehat{\mathrm{MMD}}_\mathrm{U}^2$ is an instance of a class of statistics known as U-statistics, introduced by Hoeffding (1948). We will only need (standard) U-statistics of order two here.

**Definition 2.4.** *Let $Z_1, \ldots, Z_n$ be i.i.d. random variables with support in $\mathcal{Z}$, and let $h : \mathcal{Z} \times \mathcal{Z} \to \mathbb{R}$ be symmetric in the sense that $h(z_1, z_2) = h(z_2, z_1)$. A* U-statistic of order two *is defined as*

$$U_n = \frac{1}{n(n-1)} \sum_{i \neq j} h(Z_i, Z_j).$$

The function $h$ is called the kernel of the U-statistic (not to be confused with the RKHS kernel $k$); for $\widehat{\mathrm{MMD}}_\mathrm{U}^2$ it is (2). These statistics have many general properties; we will be particularly interested in their variance.

**Proposition 2.5.** *Let $U_n$ be a U-statistic of order two with kernel $h$. Then, when $n \geq 2$,*

$$\mathrm{Var}(U_n) = \frac{4(n-2)}{n(n-1)} \mathop{\mathrm{Var}}_{Z_1} \big[ \mathop{\mathbb{E}}_{Z_2} [h(Z_1, Z_2) \mid Z_1] \big] + \frac{2}{n(n-1)} \mathop{\mathrm{Var}}_{Z_1, Z_2} [h(Z_1, Z_2)]$$

$$= \frac{4n-6}{n(n-1)} \mathop{\mathrm{Var}}_{Z_1} \big[ \mathop{\mathbb{E}}_{Z_2} [h(Z_1, Z_2) \mid Z_1] \big] + \frac{2}{n(n-1)} \mathop{\mathbb{E}}_{Z_1} \mathop{\mathrm{Var}}_{Z_2} [h(Z_1, Z_2) \mid Z_1].$$

The first form is a textbook result, shown e.g. by Lee (2019, Section 1.3) or Serfling (1980, Section 5.2). The second follows immediately from the first by the law of total variance. While this first form is well-known and can be used to give an explicit form for the variance of $\widehat{\mathrm{MMD}}_\mathrm{U}^2$ (Sutherland and Deka, 2019), we can actually simplify the form considerably. The following result, shown in Appendix B, uses the approach of He et al. (2025, Theorem 6.1).

**Proposition 2.6.** *In Setting A, we have that*

$$\mathop{\mathrm{Var}}_{\substack{\mathbf{S}_P \sim P^n \\ \mathbf{S}_Q \sim Q^n}} \left( \widehat{\mathrm{MMD}}_\mathrm{U}^2(\mathbf{S}_P, \mathbf{S}_Q) \right) = \frac{4}{n} \langle \mu_P - \mu_Q, (C_P + C_Q)(\mu_P - \mu_Q) \rangle_{\mathcal{H}} + \frac{2}{n(n-1)} \|C_P + C_Q\|_{\mathrm{HS}}^2. \tag{4}$$

For a general U-statistic of order two, $\mathrm{Var}(U_n)$ has three possible asymptotic behaviours: if $\mathrm{Var}(U_n) = \Theta(1/n)$, we call $U_n$ *non-degenerate* or zeroth-order degenerate. If $\mathrm{Var}(U_n) = \Theta(1/n^2)$, we call $U_n$ *first-order degenerate*. Otherwise, $\mathrm{Var}(U_n) = 0$, which we term *infinitely degenerate*. Proposition 2.6 allows us to almost fully characterize the degeneracy of $\widehat{\mathrm{MMD}}_\mathrm{U}^2$. One of its results needs the following stronger assumptions:

**Setting B.** *In Setting A, further assume that $\mathcal{X} = \mathbb{R}^d$, $k$ is real-analytic, $\sup_{x \in \mathcal{X}} k(x,x)$ is finite, and the supports of $P$ and $Q$ each have positive Lebesgue measure.*

In Setting B, which encompasses many common kernel choices such as the Gaussian, every function in $\mathcal{H}$ is real-analytic (Chwialkowski et al., 2015, Lemma 1). The following result is proved in Appendix C.

**Theorem 2.7.** *In Setting A, $\widehat{\mathrm{MMD}}_\mathrm{U}^2$ is infinitely degenerate (the variance is zero) if and only if $C_P = C_Q = \mathbf{0}$. Note that an infinitely degenerate MMD estimate may still be nonzero, such as if $P$ and $Q$ are distinct point masses.*

*Otherwise suppose at least one of $C_P, C_Q$ is nonzero, so $\widehat{\mathrm{MMD}}_\mathrm{U}^2$'s order of degeneracy is zero or one. Then:*

    *(i) If $\mu_P = \mu_Q$, $\widehat{\mathrm{MMD}}_\mathrm{U}^2$ is first-order degenerate.*

    *(ii) When $\mu_P \neq \mu_Q$, $\widehat{\mathrm{MMD}}_{\mathrm{U}}^2$ may be either non-degenerate or first-order degenerate.*

    *(iii) Suppose $\mathcal{X}$ is a topological space, $k(x, \cdot)$ is continuous for each $x$, and $\sup_{x \in \mathcal{X}} k(x, x)$ is finite. Further assume the supports of $P$ and $Q$ are not disjoint. Then $\widehat{\mathrm{MMD}}_{\mathrm{U}}^2$ is degenerate if and only if $\mu_P = \mu_Q$.*

    *(iv) In Setting B, $\widehat{\mathrm{MMD}}_{\mathrm{U}}^2$ is degenerate if and only if $\mu_P = \mu_Q$.*

    *(v) In Setting B, $\langle \mu_P - \mu_Q, C_P(\mu_P - \mu_Q)\rangle > 0$ if and only if $\langle \mu_P - \mu_Q, C_Q(\mu_P - \mu_Q)\rangle > 0$.*

It is well-known that if $\mu_P = \mu_Q$, the $1/n$ term is zero. To the best of our knowledge, however, it has not been previously recognized in the literature that even when $\mu_P \neq \mu_Q$, the estimator may be first-order degenerate, and several papers make (informal) claims to the contrary.[2] In many situations of interest, however, this is impossible, as shown in parts (iii) and (iv). We also note that (iv) remains true if only one of $P, Q$ has a support with positive Lebesgue measure.

The asymptotic behavior of U-statistics is also highly relevant, and determined by the degree of degeneracy. Here we need only textbook results, as shown by Lee (2019, Section 3.2) or Serfling (1980, Section 5.5).

**Theorem 2.8.** *Let $U_n$ be a U-statistic of order two with kernel $h$. Suppose that $U_n$ has mean $\theta = \mathbb{E}\, h(X_1, X_2)$, and let $\sigma_1^2 = 4\operatorname{Var}_{X_1}[\mathbb{E}_{X_2}[h(X_1, X_2) \mid X_1]]$ be the leading term in the variance decomposition.*

*It holds as $n \to \infty$ that $\sqrt{n}\,(U_n - \theta) \xrightarrow{d} \mathcal{N}(0, \sigma_1^2)$, where if $\sigma_1 = 0$ this convergence is to the point mass at $0$.*

*If $\sigma_1 = 0$, it also holds as $n \to \infty$ that $n\,(U_n - \theta) \xrightarrow{d} \sum_{j=1}^{\infty} \lambda_j(V_j^2 - 1)$, where the $\{V_j\}_{j=1}^{\infty}$ are independent standard normal variables, and $(\lambda_j)_{j=1}^{\infty}$ are the eigenvalues of the integral equation $\mathbb{E}_{X_2}[h(x_1, X_2)f(X_2)] = \lambda f(x_1)$.*

### 2.4 MMD-based tests

The eigenvalues $\lambda_j$ in the degenerate case, as they depend on the kernel and the distribution, are often difficult to find. Because $\widehat{\mathrm{MMD}}_{\mathrm{U}}^2$ is degenerate under the null hypothesis, where $\mathrm{MMD}(P, Q) = 0$, the test threshold must be set based on the more complex distribution. The eigenvalues $\lambda_j$ can be estimated based on eigendecomposition of the sample kernel matrix (Gretton et al., 2009), but it is usually faster and more effective to choose a threshold based on permutation testing (e.g. Sutherland et al., 2017). A variant of this procedure also achieves finite-sample valid test thresholds (Hemerik and Goeman, 2018), rather than the only asymptotic validity achieved from consistent estimates of $\lambda_j$.

Theorems 2.7 and 2.8 imply that if $\mu_P = \mu_Q$, then the $(1-\alpha)$th quantile of $\widehat{\mathrm{MMD}}_{\mathrm{U}}^2$ will be either $\Theta(1/\sqrt{n})$ or simply 0. On the other hand, if $\mu_P \neq \mu_Q$ so that $\mathrm{MMD}^2 > 0$, then $\widehat{\mathrm{MMD}}^2$ will be one of $\mathrm{MMD}^2 + \mathcal{O}_p(1/\sqrt{n})$, $\mathrm{MMD}^2 + \mathcal{O}_p(1/n)$, or simply $\mathrm{MMD}^2$, depending on the degeneracy behaviour. Thus as $n \to \infty$, any test whose asymptotic level is controlled will be consistent, regardless of the degree of degeneracy. That is, whenever $\mu_P \neq \mu_Q$, an MMD test will eventually reject as $n \to \infty$.

To do so, however, may require a very large number of samples when the kernel is a poor match to the problem at hand; for example, a Gaussian kernel based on image pixels does a reasonable job at identifying pixel-level shifts on simple aligned image datasets like MNIST (see Sutherland et al., 2017), but would require huge numbers of samples to identify "semantic" shifts in more complex natural image distributions.

To address this issue, Gretton, Sejdinovic, et al. (2012), Sutherland et al. (2017), Liu et al. (2020), and Kübler, Stimper, et al. (2022) maximize the asymptotic *power* of a given MMD test with equal sample sizes. Assuming that $\sigma_1(P, Q)^2$ as defined in Theorem 2.8 is nonzero,

$$\Pr\left(n\,\widehat{\mathrm{MMD}}_{\mathrm{U}}^2(\mathbf{S}_P, \mathbf{S}_Q) > c_\alpha\right) = \Pr\left(\sqrt{n}\frac{\widehat{\mathrm{MMD}}_{\mathrm{U}}^2(\mathbf{S}_P, \mathbf{S}_Q) - \mathrm{MMD}^2(P, Q)}{\sigma_1(P, Q)} > \frac{\frac{c_\alpha}{\sqrt{n}} - \sqrt{n}\,\mathrm{MMD}^2(P, Q)}{\sigma_1(P, Q)}\right),$$

---

[2]For instance, see the sentence just before (3) of Sutherland et al. (2017), the last sentence in the paragraph following (3) of Deka and Sutherland (2023), or the discussion around Theorem 1 of Kübler, Jitkrittum, et al. (2022).

When $\sigma_1 > 0$, the left-hand side of the inequality converges in distribution to a standard normal, and so

$$\Pr\left(n\,\widehat{\mathrm{MMD}}_{\mathrm{U}}^2(\mathbf{S}_P, \mathbf{S}_Q) > c_\alpha\right) \sim 1 - \Phi\left(\frac{\frac{c_\alpha}{\sqrt{n}} - \sqrt{n}\,\mathrm{MMD}^2(P,Q)}{\sigma_1(P,Q)}\right) = \Phi\left(\sqrt{n}\frac{\mathrm{MMD}^2(P,Q)}{\sigma_1(P,Q)} - \frac{c_\alpha}{\sqrt{n}\sigma_1(P,Q)}\right),$$

where $a \sim b$ denotes $a/b \to 1$ and $\Phi$ is the cdf of the standard normal distribution. Since $\mathrm{MMD}(P,Q)$, $\sigma_1(P,Q)$, and $c_\alpha$ are population quantities that do not depend on $n$, for large sample sizes the asymptotic power expression is dominated by the signal-to-noise ratio $\mathrm{MMD}^2(P,Q)/\sigma_1(P,Q)$. Thus, we can choose a kernel by maximizing a finite-sample estimate of this quantity on a training set, then run a test with that kernel on an independent test set.

## 3 Asymptotic distribution of the MMD estimate

To generalize this approach to the case where $n_X \neq n_Y$, we will derive the asymptotic distributions of the estimator $\widehat{\mathrm{MMD}}^2$, rather than $\widehat{\mathrm{MMD}}_{\mathrm{U}}^2$. To do so, we fill in results about the theory of generalized U-statistics (Serfling, 1980, Section 5.5), of which the $\widehat{\mathrm{MMD}}^2$ estimator is an instance even when $n_X \neq n_Y$.

Gretton, Borgwardt, et al. (2012, Theorem 12) previously considered the asymptotics of $\widehat{\mathrm{MMD}}^2$ when $n_X \neq n_Y$, showing that if $n_X/n_Y$ converges to a positive, finite constant and $\mathrm{MMD}(P,Q) = 0$, then $(n_X + n_Y)\,\widehat{\mathrm{MMD}}^2$ converges in distribution to a slightly different sum of shifted chi-squared variates. Our results, by contrast, will also allow $n_X/n_Y \to 0$ or $n_X/n_Y \to \infty$; to do this, instead of scaling by $n_X + n_Y$, we scale by $\min(n_X, n_Y)$. In the proportional setting, this only differs by a constant, but we also allow for non-proportional settings such as $n_Y = n_X^2$.

Before giving our results, we first empirically demonstrate in Figures 1 and 2 that $\min(n_X, n_Y)$ is indeed the correct scaling. In the proportional regime, either scaling works, while in a non-proportional setting only the $\min(n_X, n_Y)$ scaling leads to convergence; it does so to the distributions predicted by our theorems.

### 3.1 Generalized U-Statistics

To analyze $\widehat{\mathrm{MMD}}^2$, we will use the following concept.

**Definition 3.1** (Generalized U-statistic). *For $j \in [c] = \{1, 2, \ldots, c\}$, let $(X_{ij})_{i \in [n_j]} \sim \mu_j^{n_j}$ be mutually independent random variables. Let $h$ be a real-valued function of $m_1 + \cdots + m_c$ arguments which is symmetric in the sense that the value of $h(x_{11}, \ldots, x_{m_1 1}; \ldots; x_{1c}, \ldots, x_{m_c c})$ remains unchanged if we permute any block of arguments $(x_{1j}, \ldots, x_{m_j j})$. The c-sample generalized U-statistic associated with kernel $h$ is defined by*

$$U_{n_{\min}} = \prod_{j=1}^c \binom{n_j}{m_j}^{-1} \sum_{\sigma_1} \cdots \sum_{\sigma_c} h(X_{\sigma_1(1),1}, \ldots, X_{\sigma_1(m_j),1}; \ldots; X_{\sigma_c(1),k}, \ldots, X_{\sigma_c(m_c),c}), \tag{5}$$

*where $n_{\min} = \min\{n_1, \ldots, n_c\}$, and $\sigma_j$ varies over each injection from $[m_j]$ to $[n_j]$. For brevity, we may also refer to $U_n$ as a c-sample U-statistic.*

As has been previously pointed out by Kim et al. (2022) and Schrab et al. (2023), $\widehat{\mathrm{MMD}}^2$ can be viewed as a generalized U-statistic with $c = 2$, $m_1 = m_2 = 2$, using the kernel

$$h(x_i, x_j; y_i, y_j) := k(x_i, x_j) + k(y_i, y_j) - \frac{1}{2}\left[k(x_i, y_j) + k(x_j, y_i) + k(x_i, y_i) + k(x_j, y_j)\right]. \tag{6}$$

While a direct implementation of (5) would sum over $\mathcal{O}(n_X^2 n_Y^2)$ evaluations of the function $h$, in fact each term in this $h$ considers at most two elements; thus many terms in the average are irrelevant. An implementation taking this into account becomes exactly that of $\widehat{\mathrm{MMD}}^2$ in (1), with $\mathcal{O}((n_X + n_Y)^2)$ kernel evaluations.

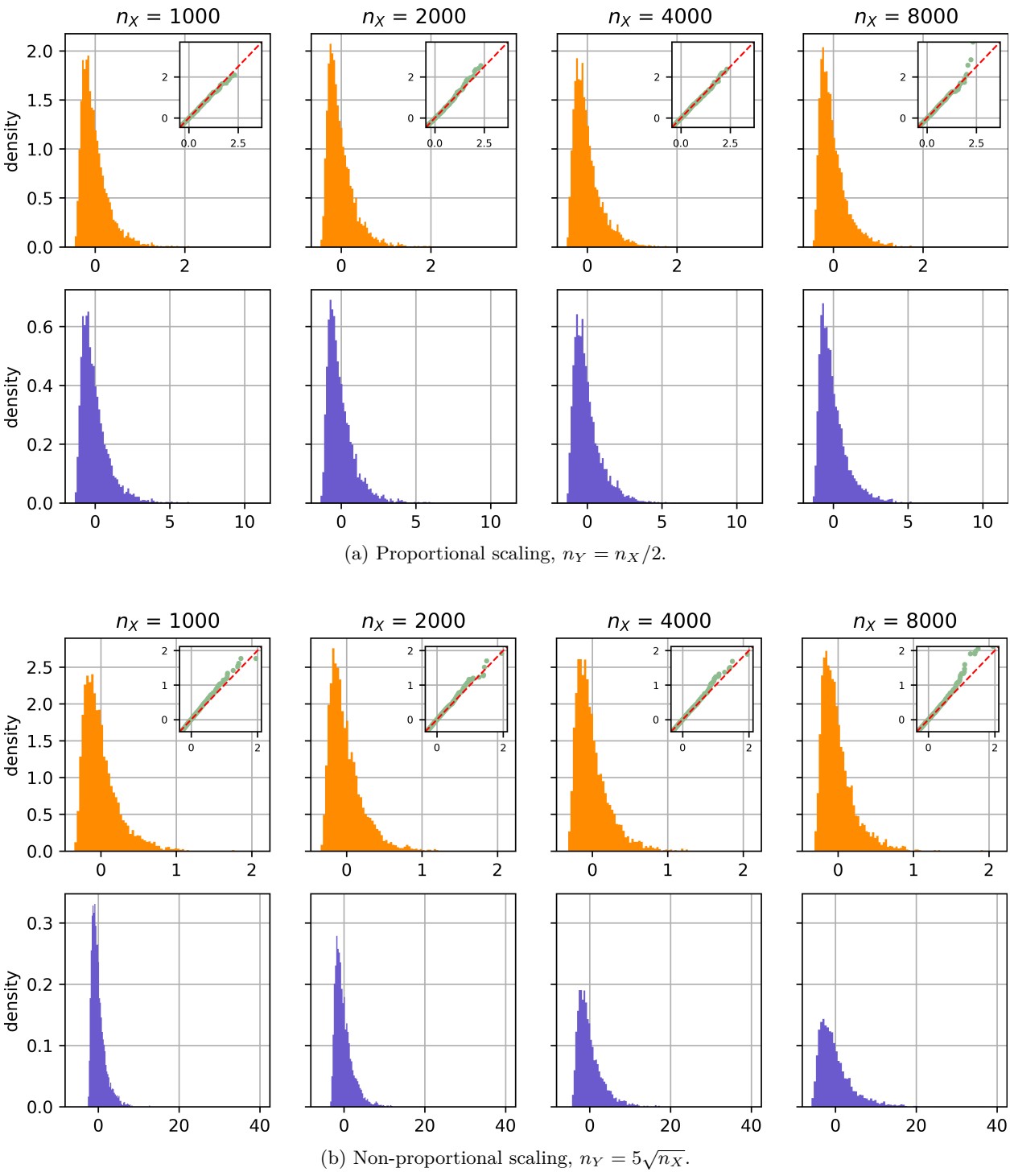

Figure 1: Histograms of $n\,\widehat{\mathrm{MMD}}^2$ for $P = Q = \mathrm{Laplace}(0, 1/\sqrt{2})$ and a unit-lengthscale Gaussian kernel; orange (top) rows use $n_{\min} = \min(n_X, n_Y)$, while blue (bottom) rows use $n_+ = n_X + n_Y$. In the proportional setting (panel a), both converge; the only difference is a constant scaling, since $n_X + n_X/2 = 3\min(n_X, n_X/2)$. In the non-proportional setting (panel b), however, it is clear that the $n_X + n_Y$ scaling is not converging in distribution, while the $\min(n_X, n_Y)$ scaling is. The $\min(n_X, n_Y)$ results include inset Q-Q plots, comparing empirical quantiles to those predicted by the limiting distributions of Theorem 3.7; eigenvalues in that distribution are estimated based on a sample with $n_X = n_Y = 5000$.

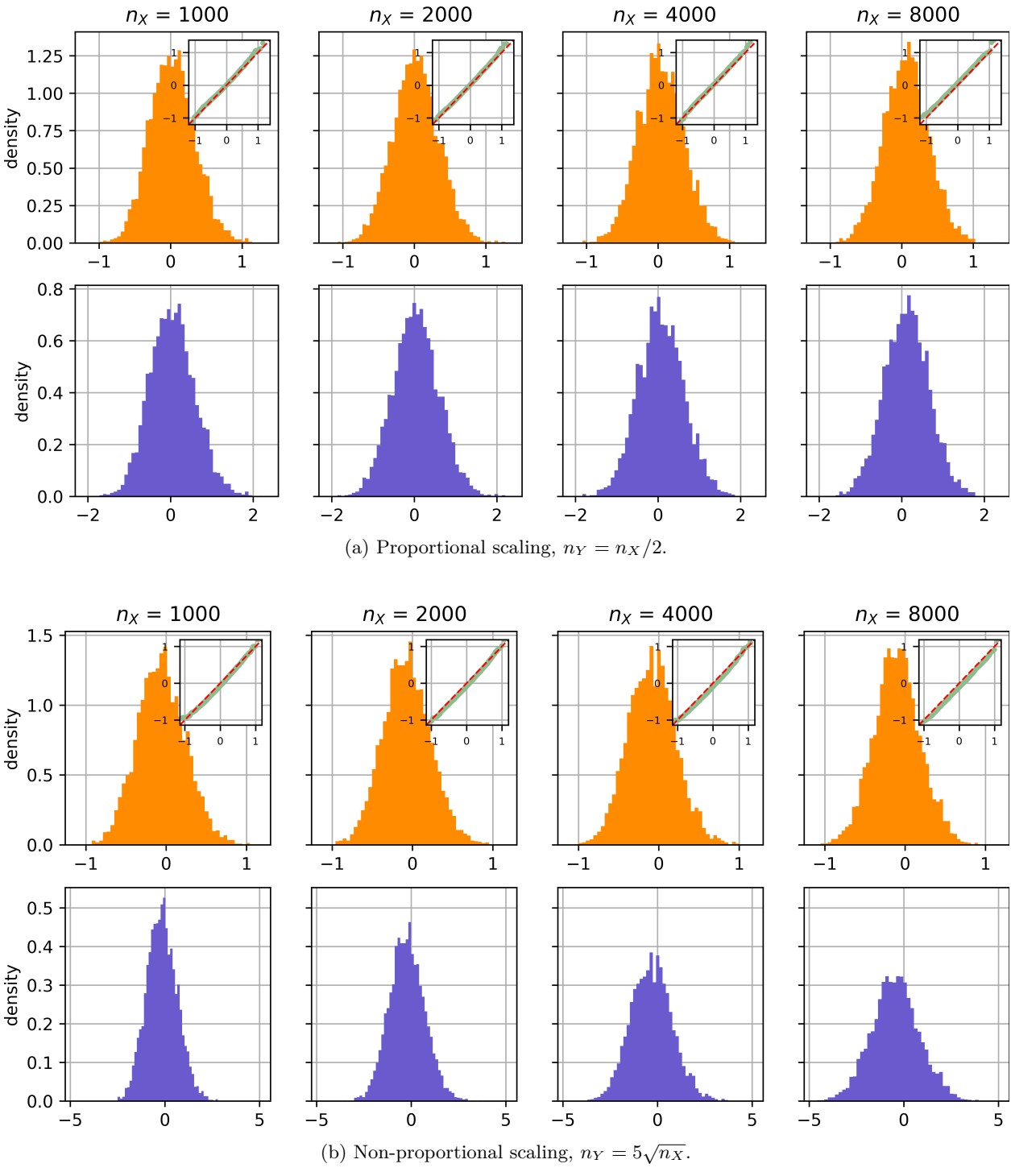

(a) Proportional scaling, $n_Y = n_X/2$.

(b) Non-proportional scaling, $n_Y = 5\sqrt{n_X}$.

Figure 2: Histograms of $\sqrt{n}(\widehat{\mathrm{MMD}}^2 - \mathrm{MMD})$ for $P = \mathrm{Laplace}(0, 1/\sqrt{2})$, $Q = \mathrm{Laplace}(0, 3)$, and $k$ is a unit-lengthscale Gaussian kernel; here $\mathrm{MMD}(P, Q) > 0$ is estimated based on 10000 samples. As in Figure 1, orange (top) rows use $n_{\min} = \min(n_X, n_Y)$, while blue (bottom) use $n_+ = n_X + n_Y$. We again see that the $n_X + n_Y$ scaling is clearly not converging in distribution in the non-proportional setting (panel b), while $\min(n_X, n_Y)$ converges in accordance with the distribution from Theorem 3.9. Quantiles of the limiting distribution are computed based on a variance estimated based on a sample with $n_X = n_Y = 10000$.

Let $\mathbf{X} = (X_{11}, \ldots, X_{m_1 1}; \ldots; X_{1c}, \ldots, X_{m_c c}) \sim \prod_{j=1}^c \mu_j^{m_j}$. Sen (1974) derived the variance of a generalized U-statistic as

$$\mathrm{Var}(U_n) = \sum_{d_1=0}^{m_1} \cdots \sum_{d_c=0}^{m_c} \left( \prod_{j=1}^c \binom{n_j}{m_j}^{-1} \binom{m_j}{d_j} \binom{n_j - m_j}{m_j - d_j} \right) \zeta_{d_1 \ldots d_c} \tag{7}$$

$$\text{where} \quad \zeta_{d_1 \ldots d_c} = \mathrm{Var}(\mathbb{E}(h(\mathbf{X}) \mid X_{11}, \ldots, X_{d_1 1}, \ldots, X_{1c}, \ldots, X_{d_c c})).$$

This motivates the following definition.

**Definition 3.2.** *We say that a generalized U-statistic $U_{n_{\min}}$ has order of degeneracy $r$ if $\zeta_{d_1 \ldots d_c} = 0$ for all $d_1 + \cdots + d_c \leq r$, and $\zeta_{d_1 \ldots d_c} > 0$ for at least one $d_1 + \cdots + d_c = r + 1$, where the $\zeta$s are as in (7). If all $\zeta_{d_1 \ldots d_c} = 0$, we say its order of degeneracy is infinite. If $U_{n_{\min}}$ has order of degeneracy zero, we say it is non-degenerate.*

The following result is then immediate from (7), since $\binom{n_j}{m_j}^{-1} \binom{m_j}{d_j} \binom{n_j - m_j}{m_j - d_j} = \Theta(n_j^{-d_j})$:

**Proposition 3.3.** *For a generalized U-statistic with finite order of degeneracy $r$,*

$$\mathrm{Var}(U_{n_{\min}}) = \sum_{d_1=0}^{m_1} \cdots \sum_{d_c=0}^{m_c} \Theta \left( \prod_{j=1}^c n_j^{-d_j} \right) \zeta_{d_1 \ldots d_c} = \mathcal{O} \left( n_{\min}^{-(r+1)} \right). \tag{8}$$

For the MMD in particular, we can express the variance in a form similar in spirit to (4).

**Theorem 3.4.** *Under Setting A, it holds for any $n_X, n_Y \geq 2$ that*

$$\begin{aligned}
\mathrm{Var}_{\substack{\mathbf{S}_P \sim P^{n_X} \\ \mathbf{S}_Q \sim Q^{n_Y}}} (\widehat{\mathrm{MMD}}^2(\mathbf{S}_P, \mathbf{S}_Q)) = {} & \frac{4}{n_X} \left( 1 - \frac{4}{n_Y} \cdot \frac{n_X - 2}{n_X - 1} \cdot \frac{n_Y - 2}{n_Y - 1} \right) \langle \mu_P - \mu_Q, C_P(\mu_P - \mu_Q) \rangle \\
& + \frac{4}{n_Y} \left( 1 - \frac{4}{n_X} \cdot \frac{n_X - 2}{n_X - 1} \cdot \frac{n_Y - 2}{n_Y - 1} \right) \langle \mu_P - \mu_Q, C_Q(\mu_P - \mu_Q) \rangle \\
& + \frac{2}{n_X(n_X - 1)} \| C_P \|_{\mathrm{HS}}^2 + \frac{2}{n_Y(n_Y - 1)} \| C_Q \|_{\mathrm{HS}}^2 + \frac{4}{n_X n_Y} \langle C_P, C_Q \rangle_{\mathrm{HS}} \\
& + \frac{16}{n_X n_Y} \left( \frac{n_X - 2}{n_X - 1} \cdot \frac{n_Y - 2}{n_Y - 1} \right) [\langle \mu_P, C_P \mu_P \rangle + \langle \mu_Q, C_Q \mu_Q \rangle].
\end{aligned}$$

Notice that each of the factors in large parentheses has limit 1 as $n_X, n_Y \to \infty$, regardless of their relative rate. The proof of Theorem 3.4, which follows from (7), is in Appendix D. The computation of conditional variances also yields the following result about degeneracy.

**Proposition 3.5.** *$\widehat{\mathrm{MMD}}^2$ has the same order of degeneracy as $\widehat{\mathrm{MMD}_{\mathrm{U}}}^2$; thus Theorem 2.7 applies to $\widehat{\mathrm{MMD}}^2$ as well.*

*Remark* 3.6. In the proportional regime $n_i = \Theta(n_{\min})$, the rate of Proposition 3.3 becomes $\Theta(n_{\min}^{-(r+1)})$, i.e. the variance characterization is tight for any generalized U-statistic. The same holds for $\widehat{\mathrm{MMD}}^2$ in Setting B via Theorem 2.7 part (v). In general, however, we only have $\mathcal{O}(n_{\min}^{-(r+1)})$, not $\Theta$: for instance, if $C_P = 0$, $\langle \mu_P - \mu_Q, C_Q(\mu_P - \mu_Q) \rangle > 0$, and $\langle \mu_Q, C_Q \mu_Q \rangle > 0$, $\widehat{\mathrm{MMD}}^2$ is non-degenerate but its variance in Equation (4) is $\Theta \left( 0 + \frac{1}{n_Y} + 0 + \frac{1}{n_Y^2} + 0 + \frac{1}{n_X n_Y} \right) = \Theta \left( \frac{1}{n_Y} \right)$, which for $n_Y = n_X^{10}$ is $\Theta(n_{\min}^{-10})$, not $\Theta(n_{\min}^{-1})$.

## 3.2 Distribution when the MMD is zero

We first consider the distribution of the estimator when $\mu_P = \mu_Q$, which implies first-order degeneracy. (Per Theorem 2.7, first-order degeneracy is also possible when $\mu_P \neq \mu_Q$, but relatively uncommon; since it makes the proof more difficult, we do not handle that case here.)

In Theorem E.5, stated in Appendix E, we derive the asymptotic distribution of a general U-statistic with first order degeneracy and a square-integrable kernel. To our knowledge, this result (in its treatment of the

case $c > 2$ and arbitrary $m_j$) is a substantial generalization of classical theorems. Our proof extends the approach of Serfling (1980, Section 5.5.2) and Anderson et al. (1994), which was later used specifically for $\widehat{\mathrm{MMD}}^2$ in the proportional setting by Gretton, Borgwardt, et al. (2012, Theorem 12). In the degenerate case, we also treat asymmetric kernels in a shared manner to Koroljuk and Borovskich (1993, theorem 4.5.3), with the use of an orthogonal expansion (Lemma E.4).

Since our assumptions are minimal, the resulting distribution is somewhat complicated to describe. As such, we leave the details to Appendix E, and only give the particular case for $\widehat{\mathrm{MMD}}^2$ here.

**Theorem 3.7.** *Assume Setting A and that* $\mathrm{MMD}^2(P, Q) = 0$. *Assume* $\min\{n_X, n_Y\}/n_X \to \rho_X$ *and* $\min\{n_X, n_Y\}/n_Y \to \rho_Y$ *for some* $\rho_X, \rho_Y$ *in* $[0, 1]$. $\widehat{\mathrm{MMD}}^2$ *converges in distribution as*

$$\min\{n_X, n_Y\} \widehat{\mathrm{MMD}}^2 \xrightarrow{d} (\rho_X + \rho_Y) \sum_{l=1}^{\infty} \lambda_l \left( Z_l^2 - 1 \right),$$

*where each* $Z_l$ *is independently* $\mathcal{N}(0, 1)$, *and the* $\lambda_l$ *are the eigenvalues of the integral equation*

$$\mathop{\mathbb{E}}_{X} \left[ \langle \phi(X) - \mu_P, \phi(y) - \mu_P \rangle g(X) \right] = \lambda g(y). \tag{9}$$

That is, $\min\{n_X, n_Y\} \widehat{\mathrm{MMD}}^2$ converges in distribution to an infinite mixture of centered chi-squared distributions, where the mixture components are distribution- and kernel-dependent.

For comparison's sake, in our notation Theorem 12 of Gretton, Borgwardt, et al. (2012) says that[3] if $n_X/(n_X + n_Y) \to \ell_X$ and $n_Y/(n_X + n_Y) \to \ell_Y = 1 - \ell_X$ for $\ell_X, \ell_Y \in (0, 1)$, then

$$(n_X + n_Y) \widehat{\mathrm{MMD}}^2 \xrightarrow{d} \frac{1}{\ell_X \ell_Y} \sum_{l=1}^{\infty} \lambda_l \left( Z_l^2 - 1 \right).$$

To confirm our results agree, suppose $n_Y = \frac{1}{\rho_Y} n_X$, with $\rho_Y \in (0, 1]$, so $\rho_X = 1$. Thus $\ell_X = \frac{1}{1 + \frac{1}{\rho_Y}} = \frac{\rho_Y}{1 + \rho_Y}$ and $\ell_Y = \frac{1}{1 + \rho_Y}$. Let $D = \sum_l \lambda_l (Z_l^2 - 1)$. Then our result is $n_X \widehat{\mathrm{MMD}}^2 \xrightarrow{d} (1 + \rho_Y) D = \frac{1}{\ell_Y} D$, while since $n_X + n_Y = (1 + \frac{1}{\rho_Y}) n_X = \frac{1}{\ell_X} n_X$, theirs equivalently says that $\frac{1}{\ell_X} n_X \widehat{\mathrm{MMD}}^2 \xrightarrow{d} \frac{1}{\ell_X \ell_Y} D$. As demonstrated by Figures 1 and 2, however, their result cannot be generalized to non-proportional asymptotics.

### 3.3 Non-degenerate asymptotic normality

Our derivation of the asymptotic distribution under the alternative follows from a result on generalized U-statistics, which is simpler to state than the corresponding result for the degenerate case. The proceeding theorem follows readily from existing literature, with a similar result being presented as an exercise, without solution, by Serfling (1980, section 5.5.1) as well as in Koroljuk and Borovskich (1993, theorem 4.5.1). In more recent times, the asymptotic normality of generalized U-statistics has been considered for product form estimators (Kuntz et al., 2022) although these results specialize to the case $n_1 = \cdots = n_c$.

**Theorem 3.8.** *Let* $U_{n_{\min}}$ *be a* $c$-*sample U-statistic with kernel* $h$, *where* $n_{\min} = \min\{n_1, \ldots, n_c\}$. *If* $n_{\min}/n_j \to \rho_j \in [0, 1]$ *for each* $j \in [c]$, *then*

$$\sqrt{n_{\min}}(U_{n_{\min}} - \mathbb{E}[h(\mathbf{X})]) \xrightarrow{d} \mathcal{N}\left( 0, \sum_{j=1}^{c} \rho_j m_j^2 \operatorname{Var}\left( \mathbb{E}\left[ h(\mathbf{X}) \mid X_{1j} \right] \right) \right),$$

*where* $\mathcal{N}(0, 0)$ *is interpreted as a point mass at 0. In particular, if* $U_{n_{\min}}$ *is non-degenerate and each* $\rho_j > 0$ *(the proportional regime), the distribution above is normal with positive variance.*

---

[3]They wrote their limiting distribution as $\sum_{l=1}^{\infty} \lambda_l \left( (\ell_X^{-1/2} A_l - \ell_Y^{-1/2} B_l)^2 - (\ell_X \ell_Y)^{-1} \right)$ for $A_l, B_l$ standard normal; this implies the above result by noting that $\ell_X^{-1/2} A_l - \ell_Y^{-1/2} B_l \overset{d}{=} \sqrt{\frac{1}{\ell_X} + \frac{1}{\ell_Y}} Z_l = \sqrt{\frac{\ell_Y + \ell_X}{\ell_X \ell_Y}} Z_l = \sqrt{\frac{1}{\ell_X \ell_Y}} Z_l$.

The following application to MMD is immediate from Theorem 3.8 and Theorem 2.7 part (v).

**Theorem 3.9.** *Assume $\min\{n_X, n_Y\}/n_X \to \rho_X$ and $\min\{n_X, n_Y\}/n_Y \to \rho_Y$ for some $\rho_X, \rho_Y \in [0,1]$. The estimator $\widehat{\mathrm{MMD}}^2$ is asymptotically normal, following*

$$\sqrt{\min\{n_X, n_Y\}}(\widehat{\mathrm{MMD}}^2 - \mathrm{MMD}^2) \xrightarrow{d} \mathcal{N}(0, 4\rho_X \zeta_X + 4\rho_Y \zeta_Y),$$

*where $\zeta_X = \langle \mu_P - \mu_Q, C_P(\mu_P - \mu_Q) \rangle$ and $\zeta_Y = \langle \mu_P - \mu_Q, C_Q(\mu_P - \mu_Q) \rangle$.*

*In Setting B, when $\widehat{\mathrm{MMD}}^2$ is non-degenerate the distribution above has positive variance.*

**Power maximization**

Paralleling the case for $\widehat{\mathrm{MMD}_{\mathrm{U}}}^2$, Theorem 3.9 implies that when $\sigma^2 = 4\rho_X \zeta_X + 4\rho_Y \zeta_Y > 0$ and $n_{\min} = \min\{n_X, n_Y\}$,

$$\Pr\left(n_{\min} \widehat{\mathrm{MMD}}^2(\mathbf{S}_P, \mathbf{S}_Q) > c_\alpha\right) \sim \Phi\left(\sqrt{n_{\min}}\frac{\mathrm{MMD}^2}{\sigma} - \frac{c_\alpha}{\sqrt{n_{\min}}\,\sigma}\right).$$

Thus we can choose the asymptotically best kernel for a test with a given sample size ratio by maximizing the signal-to-noise ratio $\mathrm{MMD}^2/\sigma$, which we can estimate by $\widehat{\mathrm{MMD}}^2/(\hat{\sigma} + \lambda)$ for some small $\lambda > 0$.

This argument does not apply when the variance $\sigma$ of Section 3.3 is zero, or equivalently when $\widehat{\mathrm{MMD}}^2$ is degenerate. This will always be the case when the true population MMD is zero, but this is not a concern, as under this null hypothesis permutation testing will ensure our test behaves appropriately regardless of the selected kernel. It remains to wonder, however, what might happen when $\widehat{\mathrm{MMD}}^2$ is degenerate but $\mathrm{MMD} > 0$, as in Theorem 2.7 part (ii). If our optimization has come across such a kernel, the true signal-to-noise ratio will be infinite (a positive numerator, a zero denominator), and in fact the power of a test with this kernel should be excellent. The estimated signal-to-noise will also be extremely large, since it will estimate a high numerator and (after regularization) a very small denominator. Thus, while we do not have a full theoretical understanding of the asymptotics of this situation, it is not a problem for power maximization.

**Estimating the variance**

To find an estimator $\hat{\sigma}$, notice that

$$\zeta_X = \mathop{\mathrm{Var}}_X\big(\mathop{\mathbb{E}}_{X'}[k(X, X')]\big) + \mathop{\mathrm{Var}}_X\big(\mathop{\mathbb{E}}_Y[k(X, Y)]\big) - 2\mathop{\mathrm{Cov}}_X\big(\mathop{\mathbb{E}}_{X'}[k(X, X')], \mathop{\mathbb{E}}_Y[k(X, Y)]\big)$$

$$\zeta_Y = \mathop{\mathrm{Var}}_Y\big(\mathop{\mathbb{E}}_{Y'}[k(Y, Y')]\big) + \mathop{\mathrm{Var}}_Y\big(\mathop{\mathbb{E}}_X[k(X, Y)]\big) - 2\mathop{\mathrm{Cov}}_Y\big(\mathop{\mathbb{E}}_{Y'}[k(Y, Y')], \mathop{\mathbb{E}}_X[k(X, Y)]\big).$$

We use simple plug-in estimators of the following form:

$$\mathop{\mathrm{Var}}_X[\mathop{\mathbb{E}}_{X'}[k(X, X')]] \approx \frac{1}{n_X}\sum_{j=1}^{n_X}\left(\frac{1}{n_X}\sum_{i=1}^{n_X}k(x_i, x_j)\right)^2 - \left(\frac{1}{n_X^2}\sum_{j=1}^{n_X}\sum_{i=1}^{n_X}k(x_i, x_j)\right)^2 \tag{10}$$

$$\mathop{\mathrm{Var}}_X(\mathop{\mathbb{E}}_Y[k(X, Y)]) \approx \frac{1}{n_X}\sum_{i=1}^{n_X}\left(\frac{1}{n_Y}\sum_{j=1}^{n_Y}k(x_i, y_j)\right)^2 - \left(\frac{1}{n_X n_Y}\sum_{j=1}^{n_Y}\sum_{i=1}^{n_X}k(x_i, y_j)\right)^2 \tag{11}$$

$$\mathop{\mathrm{Cov}}_X(\mathop{\mathbb{E}}_{X'}[k(X, X')], \mathop{\mathbb{E}}_Y[k(X, Y)]) \approx \frac{1}{n_X}\sum_{i=1}^{n_X}\left(\frac{1}{n_X}\sum_{a=1}^{n_X}k(x_i, x_a)\right)\left(\frac{1}{n_Y}\sum_{b=1}^{n_Y}k(x_i, y_b)\right)$$
$$- \left(\frac{1}{n_X^2}\sum_{i=1}^{n_X}\sum_{a=1}^{n_X}k(x_i, x_a)\right)\left(\frac{1}{n_X n_Y}\sum_{j=1}^{n_X}\sum_{b=1}^{n_Y}k(x_j, y_b)\right), \tag{12}$$

giving $\hat{\zeta}_X$ as (10) + (11) − 2 · (12). The estimator $\hat{\zeta}_Y$ is analogous, and finally $\hat{\sigma} = 2\sqrt{\rho_X \hat{\zeta}_X + \rho_Y \hat{\zeta}_Y}$.

These terms, as various means of the $X$-to-$X$, $X$-to-$Y$, and $Y$-to-$Y$ kernel matrices, can be written straightforwardly in modern automatic differentiation libraries. Thus $\widehat{\text{MMD}}^2/(\hat\sigma + \lambda)$ can be easily written as an objective function for gradient-based optimization of kernel parameters. These estimators are very similar to the variance estimator used by Liu et al. (2020) and are based on the same sums of kernel terms, simply scaled in a slightly different way; thus the difficulty of implementation and computational cost are essentially the same as that of Liu et al. (2020).

These plug-in estimators are biased; unbiased estimators (perhaps also incorporating the lower-order terms of Theorem 3.4), could also be derived as was done for $\widehat{\text{MMD}}_{\text{U}}{}^2$ by Sutherland and Deka (2019). In the setting of SNR maximization, however, an unbiased estimator of $\sigma^2$ does *not* imply an unbiased estimator of $\frac{1}{\sigma}$; in fact, no such estimator can exist (Deka and Sutherland, 2023, Proposition 1). Deka and Sutherland (2023) also report that the biased estimator worked better in their experiments in a closely related setting.

The uniform convergence result of Liu et al. (2020) for the signal-to-noise ratio estimator of $\widehat{\text{MMD}}_{\text{U}}{}^2$ readily generalizes to this estimator, showing that optimizing this signal-to-noise ratio estimator does indeed optimize the true signal-to-noise ratio for $\widehat{\text{MMD}}^2$. We give an informal summary here, but in fact all results from Liu et al. (2020) – in particular their Theorem 6, Propositions 7 to 9, and the formal versions in their appendix Theorem 11 and Corollaries 12 and 13 – exactly apply when their sample size $n$ is replaced by the harmonic mean $n_{\text{harm}} = 2/(1/n_X + 1/n_Y) \geq n_{\text{min}}$. Details are given in Appendix F.

**Theorem 3.10** (Informal). *Consider a smoothly parameterized set of kernels $k_\omega$ lying in a finite-dimensional Banach space. Then, with appropriate regularization $\lambda$, $\widehat{\text{MMD}}^2/(\hat\sigma + \lambda)$ converges uniformly to $\widehat{\text{MMD}}/\sigma$ over bounded sets of parameters with variance bounded away from zero. Thus the maximizer of the estimate approaches the maximizer of the population signal-to-noise ratio.*

**Scaling with the harmonic mean**

To conclude this section, we consider an alternative scaling to $n_{\text{min}}$ which reveals more explicit comparisons between $\widehat{\text{MMD}}^2$ and $\widehat{\text{MMD}}_{\text{U}}{}^2$. Assuming that the appropriate limits exist, we have

$$n_{\text{harm}} = \frac{2}{1/n_X + 1/n_Y} = \frac{2n_{\text{min}}}{n_{\text{min}}/n_x + n_{\text{min}}/n_x} \sim \frac{2n_{\text{min}}}{\rho_X + \rho_Y}.$$

The above directly leads the following corollary to Theorem 3.9 and Theorem 3.7.

**Corollary 3.11.** *Under the conditions of Theorem 3.9, we have*

$$n_{\text{harm}} \widehat{\text{MMD}}^2 \xrightarrow{d} 2 \sum_{l=1}^{\infty} \lambda_l \left( Z_l^2 - 1 \right) \tag{13}$$

*where the $Z_l$ are independently $\mathcal{N}(0,1)$ and the $\lambda_l$ are the eigenvalues of the integral equation (9). Under the conditions of Theorem 3.7, we have*

$$\sqrt{n_{\text{harm}}}(\widehat{\text{MMD}}^2 - \text{MMD}^2) \xrightarrow{d} \mathcal{N}\left(0, 8\left[\frac{\rho_X}{\rho_X + \rho_Y}\zeta_X + \frac{\rho_Y}{\rho_X + \rho_Y}\zeta_Y\right]\right) \tag{14}$$

*where $\zeta_X = \langle \mu_P - \mu_Q, C_P(\mu_P - \mu_Q)\rangle$ and $\zeta_Y = \langle \mu_P - \mu_Q, C_Q(\mu_P - \mu_Q)\rangle$.*

Consider the hypothetical scenario where samples from $P$ are limited to $n_X$ points, though $Q$ can be easily sampled. Under the null, Gretton et al. (2008, Theorem 19) calculated that $n_X \widehat{\text{MMD}}_{\text{U}}{}^2 \xrightarrow{d} 2\sum_{l=1}^{\infty} \lambda_l \left( Z_l^2 - 1\right)$, with $\lambda_l, Z_l$ defined as in (13). In terms of variance this means that $\text{Var}(\widehat{\text{MMD}}_{\text{U}}{}^2)/\text{Var}(\widehat{\text{MMD}}^2) \approx n_{\text{harm}}/n_X = 2/(1 + 1/n_Y)$. Thus maximally exploiting an excess of samples from $Q$ (taking $n_Y \to \infty$) shrinks the variance of the sampling distribution up to a factor of two when using the estimator $\widehat{\text{MMD}}^2$. While a factor of two might seem modest, the upcoming experiments demonstrate that two-sample testing can benefit immensely from this effect.

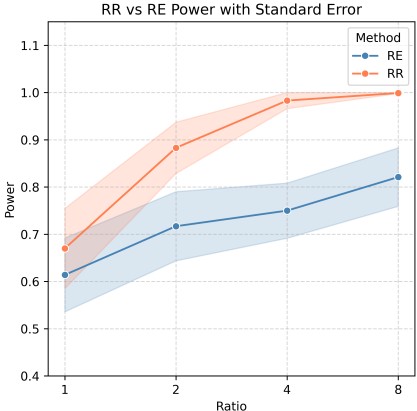

| Method | $r = 1$ | $r = 2$ | $r = 4$ | $r = 8$ |
|---|---|---|---|---|
| RE | $0.614 \pm 0.074$ | $0.717 \pm 0.069$ | $0.75 \pm 0.055$ | $0.821 \pm 0.058$ |
| RR | $0.67 \pm 0.080$ | $0.883 \pm 0.051$ | $0.983 \pm 0.016$ | $0.999 \pm 0.00094$ |

Figure 3: Comparison of RE and RR when training on 1000 samples from *CIFAR-10.1* and $r * 1000$ samples from *CIFAR-10*. The columns below $r = 1, 2, 4, 8$ contain the observed power $\pm$ standard error estimated from 10 separate rounds of training. We use $\lambda = 1 \times 10^{-8}$ for the regularization parameter in both tests. At each sample size we see that RR outperforms RE, with significantly lower variance as $r$ becomes large. These improvement are despite both procedures having access to the same amount of data during training.

## 4 Experimental comparison of tests

In this section, we evaluate the performance of two-sample tests based on our estimators when $n_X \neq n_Y$, showing that power is improved when we use as many samples as are available even when that number is asymmetric. We use a variant of permutation tests such that the Type-I error rate is exactly controlled under the null (Hemerik and Goeman, 2018).

We compare *CIFAR-10* (Krizhevsky, 2009), consisting of 60,000 images, with the *CIFAR-10.1* dataset (Recht et al., 2019) of 2,031 images. Following the experimental setup of Liu et al. (2020), we extend the framework to accommodate unequal sample sizes, taking advantage of the flexibility provided by our proposed method. We study the following procedures to compare the test power of $\widehat{\mathrm{MMD}_{\mathrm{U}}}^2$ and $\widehat{\mathrm{MMD}}^2$:

- **Ratio/equal (RE):** Extending Liu et al. (2020)'s experiment, we train on 1000 samples from *CIFAR-10.1*, but select $1000\,r$ training samples from *CIFAR-10*. Learning a deep kernel by maximizing $\widehat{\mathrm{MMD}_{\mathrm{U}}}^2 / (\hat{\sigma}_1 + \lambda)$ remains possible through mini-batching images from both image sets in equal proportions. Finally the power is estimated through 100 runs of a permutation test using $\widehat{\mathrm{MMD}_{\mathrm{U}}}^2$ on the remaining data.

- **Ratio/ratio (RR):** The training set selection is identical to RE. For training, we construct mini-batches with the proportion of *CIFAR-10.1* to *CIFAR-10* samples being $1/r$ as well. Correspondingly, we maximize the objective $\widehat{\mathrm{MMD}}^2 / (\hat{\sigma} + \lambda)$ using with $\rho_X, \rho_Y$ chosen to reflect the mini-batch proportions. The power is estimated through 100 runs of a permutation test using $\widehat{\mathrm{MMD}}^2$, again taking the proportion of the image sets to be $1/r$.

In both RE and RR, we parametrize our deep kernel using the same CNN architecture as in Liu et al. (2020) and maximize the objectives using the Adam optimizer (Kingma and Ba, 2017). Results are shown in Figure 3; we see improvement both from using more test data as well as training using our variance criterion.

## 5 Conclusion

In this paper we have shown that the generalized U-statistic estimator $\widehat{\mathrm{MMD}}^2$ provides an effective framework for two-sample testing with unequal sample sizes. $\widehat{\mathrm{MMD}}^2$ is compatible with the classical setting where the samples sizes are proportional, but our results extend to the situation where one sample is asymptotically dominant by changing the scaling from $n_X + n_Y$ to $\min\{n_X, n_Y\}$. To derive the asymptotic distributions of $\widehat{\mathrm{MMD}}^2$, we characterized generalized U-statistics based on the degeneracy of their kernel and matched this order of degeneracy with the hypotheses $H_0$ and $H_1$ under appropriate assumptions. Even when $n_X = n_Y$, while $\widehat{\mathrm{MMD}}^2$ and $\widehat{\mathrm{MMD}_\mathrm{U}}^2$ are asymptotically equivalent in probability, $\widehat{\mathrm{MMD}}^2$ has strictly lower variance on finite samples if $\langle C_P, C_Q \rangle \neq 0$; when $n_X \neq n_Y$, the variance gap can be large.

Leveraging our theoretical results, we derived an approximation to the asymptotic test power, which is (nearly) monotonic in the signal-to-noise ratio $\mathrm{MMD}^2 / \sigma$. This yields a power-optimization scheme based on maximizing $\widehat{\mathrm{MMD}}^2 / (\hat{\sigma} + \lambda)$, with which we demonstrated improved performance in the challenging problem of distinguishing CIFAR-10.1 from CIFAR-10.

The main theoretical aspect we did not fully characterize is determining the conditions under which $\zeta_X$ and $\zeta_Y$ are positive, or have matching signs. Theorem 2.7 makes significant progress towards this, in the case of continuous kernels (with overlapping support) and/or analytic kernels (with potentially non-overlapping support); typical kernels defined by neural networks will be continuous but not analytic, leaving the disjoint-support question with such kernels unknown. These questions, however, seem to be of particular interest only when the supports are very complex; distributions with "simple" disjoint supports are easy to distinguish.

### Acknowledgements

The authors would like to thank Antonin Schrab and Arthur Gretton for productive discussions.

This work was enabled by part by supported provided by the Natural Sciences and Engineering Research Council of Canada, the Canada CIFAR AI Chairs Program, Calcul Québec, the BC DRI Group, and the Digital Research Alliance of Canada.

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

# A  Proof of Theorem 2.3

**Theorem 2.3.** *When the number of samples from $P$ and $Q$ are the same, we have that*

$$\Pr_{\substack{X_1,\dots,X_n\sim P \\ Y_1,\dots,Y_n\sim Q}}\left(\left|\widehat{\mathrm{MMD}}^2-\widehat{\mathrm{MMD}}_{\mathrm{U}}^2\right|\le\frac{8\sup_{x\in\mathcal{X}}k(x,x)}{n^{3/2}}\sqrt{\log\frac{2}{\delta}}\right)\ge 1-\delta.$$

*Proof.* Letting $K=\sup_{x\in\mathcal{X}}k(x,x)$, we have that

$$k(x,y)=\langle k(x,\cdot),k(y,\cdot)\rangle\le\|k(x,\cdot)\|\|k(y,\cdot)\|=\sqrt{k(x,x)k(y,y)}\le K.$$

The difference between the estimators is

$$\begin{aligned}
\widehat{\mathrm{MMD}}^2-\widehat{\mathrm{MMD}}_{\mathrm{U}}^2 &= -\frac{2}{n^2}\sum_{i,j}k(x_i,y_j)+\frac{2}{n(n-1)}\sum_{i\ne j}k(x_i,y_j)\\
&= -\frac{2}{n^2}\sum_{i=1}^n k(x_i,y_i)+\frac{2}{n}\left(\frac{1}{n-1}-\frac{1}{n}\right)\sum_{i\ne j}k(x_i,y_j)\\
&= \frac{2}{n}\left(-\frac{1}{n}\sum_{i=1}^n k(x_i,y_i)+\frac{1}{n(n-1)}\sum_{i\ne j}k(x_i,y_j)\right).
\end{aligned}$$

This sum satisfies bounded differences: if we consider changing a single $x_i$, it changes a single term in the first sum by at most $2K$, meaning that the first average changes by at most $2K/n$. At most $n-1$ terms change in the second sum, again each by up to $2K$, giving again a total change of at most $2K/n$. Thus the overall difference in estimators changes by at most $8K/n^2$. The same holds for changing a single $y_i$. This difference is also mean zero, so since all of the $2n$ arguments $x_i$, $y_i$ are mutually independent, we can apply McDiarmid's inequality to get

$$\Pr\left(\left|\widehat{\mathrm{MMD}}^2-\widehat{\mathrm{MMD}}_{\mathrm{U}}^2\right|\le\frac{8K}{n^2}\sqrt{\frac{1}{2}(2n)\log\frac{2}{\delta}}\right)\ge 1-\delta. \qquad\square$$

# B  Proof of U-statistic variance decomposition (Proposition 2.6)

**Proposition 2.6.** *In Setting A, we have that*

$$\operatorname*{Var}_{\substack{\mathbf{S}_P\sim P^n \\ \mathbf{S}_Q\sim Q^n}}\left(\widehat{\mathrm{MMD}}_{\mathrm{U}}^2(\mathbf{S}_P,\mathbf{S}_Q)\right)=\frac{4}{n}\langle\mu_P-\mu_Q,(C_P+C_Q)(\mu_P-\mu_Q)\rangle_{\mathcal{H}}+\frac{2}{n(n-1)}\|C_P+C_Q\|_{\mathrm{HS}}^2. \tag{4}$$

*Proof.* This result follows directly from Theorem 6.1 of He et al. (2025), but we reproduce their approach here for completeness.

Define $\delta_{xy}=k(x,\cdot)-k(y,\cdot)\in\mathcal{H}$, for any $x,y\in\mathcal{X}$. Then $h$ of (2) is $h((x,y),(x',y'))=\langle\delta_{xy},\delta_{x'y'}\rangle_{\mathcal{H}}$. We also have that $\mathbb{E}\,\delta_{XY}=\mathbb{E}\,k(X,\cdot)-\mathbb{E}\,k(Y,\cdot)=\mu_P-\mu_Q$ and

$$\begin{aligned}
\mathbb{E}\,\delta_{XY}\otimes\delta_{XY} &= \mathbb{E}\,k(X,\cdot)\otimes k(X,\cdot)-\mathbb{E}\,k(X,\cdot)\otimes k(Y,\cdot)-\mathbb{E}\,k(Y,\cdot)\otimes k(X,\cdot)+\mathbb{E}\,k(Y,\cdot)\otimes k(Y,\cdot)\\
&= (C_P+\mu_P\otimes\mu_P)-\mu_P\otimes\mu_Q-\mu_Q\otimes\mu_P+(C_Q+\mu_Q\otimes\mu_Q)\\
&= C_P+C_Q+(\mu_P-\mu_Q)\otimes(\mu_P-\mu_Q).
\end{aligned}$$

For the first term in the variance decomposition of Proposition 2.5,

$$\mathbb{E}_{X',Y'}[h((x,y),(X',Y'))]=\langle\delta_{xy},\mu_P-\mu_Q\rangle_{\mathcal{H}}$$

$$\operatorname*{Var}_{X,Y}\left[\mathbb{E}_{X',Y'}[h((X,Y),(X',Y'))\mid X,Y]\right]=\langle\mu_P-\mu_Q,(C_P+C_Q)(\mu_P-\mu_Q)\rangle_{\mathcal{H}}.$$

For the second term,

$$
\begin{aligned}
\operatorname*{Var}_{X',Y'}[h((x,y),(X',Y'))] &= \operatorname*{Var}_{X',Y'}[\langle \delta_{xy}, \delta_{X'Y'}\rangle] \\
&= \mathbb{E}\left[\langle \delta_{xy}, \delta_{X'Y'}\rangle^2\right] - (\mathbb{E}\langle \delta_{xy}, \delta_{X'Y'}\rangle)^2 \\
&= \mathbb{E}\langle \delta_{xy}, \delta_{X'Y'}\rangle\langle \delta_{X'Y'}, \delta_{xy}\rangle - \langle \delta_{xy}, \mu_P - \mu_Q\rangle^2 \\
&= \mathbb{E}\langle \delta_{xy}, (\delta_{X'Y'} \otimes \delta_{X'Y'})\delta_{xy}\rangle - \langle \delta_{xy}, ((\mu_P - \mu_Q) \otimes (\mu_P - \mu_Q))\delta_{xy}\rangle \\
&= \langle \delta_{xy}, \left(\mathbb{E}(\delta_{X'Y'} \otimes \delta_{X'Y'}) - (\mu_P - \mu_Q) \otimes (\mu_P - \mu_Q)\right)\delta_{xy}\rangle \\
&= \langle \delta_{xy}, (C_P + C_Q)\delta_{xy}\rangle \\
&= \langle \delta_{xy} \otimes \delta_{xy}, C_P + C_Q\rangle_{\mathrm{HS}} \\
\mathbb{E}_{X,Y}\left[\operatorname*{Var}_{X',Y'}[h((X,Y),(X',Y')) \mid X,Y]\right] &= \left\langle \mathbb{E}_{X,Y}[\delta_{XY} \otimes \delta_{XY}], C_P + C_Q\right\rangle_{\mathrm{HS}} \\
&= \|C_P + C_Q\|_{\mathrm{HS}}^2 + \langle (\mu_P - \mu_Q) \otimes (\mu_P - \mu_Q), C_P + C_Q\rangle_{\mathrm{HS}} \\
&= \|C_P + C_Q\|_{\mathrm{HS}}^2 + \langle \mu_P - \mu_Q, (C_P + C_Q)(\mu_P - \mu_Q)\rangle_{\mathcal{H}}.
\end{aligned}
$$

Letting $\nu = \langle \mu_P - \mu_Q, (C_P + C_Q)(\mu_P - \mu_Q)\rangle_{\mathcal{H}}$ for brevity, we have obtained (4):

$$
\operatorname{Var}\left(\widehat{\mathrm{MMD}}_{\mathrm{U}}{}^2\right) = \frac{4n-6}{n(n-1)}\nu + \frac{2}{n(n-1)}\left(\|C_P + C_Q\|_{\mathrm{HS}}^2 + \nu\right) = \frac{4}{n}\nu + \frac{2}{n(n-1)}\|C_P + C_Q\|_{\mathrm{HS}}^2. \qquad \square
$$

## C  Proof of degeneracy characterization (Theorem 2.7)

**Theorem 2.7.** *In Setting A, $\widehat{\mathrm{MMD}}_{\mathrm{U}}{}^2$ is infinitely degenerate (the variance is zero) if and only if $C_P = C_Q = \mathbf{0}$. Note that an infinitely degenerate MMD estimate may still be nonzero, such as if $P$ and $Q$ are distinct point masses.*

*Otherwise suppose at least one of $C_P, C_Q$ is nonzero, so $\widehat{\mathrm{MMD}}_{\mathrm{U}}{}^2$'s order of degeneracy is zero or one. Then:*

   *(i) If $\mu_P = \mu_Q$, $\widehat{\mathrm{MMD}}_{\mathrm{U}}{}^2$ is first-order degenerate.*

   *(ii) When $\mu_P \neq \mu_Q$, $\widehat{\mathrm{MMD}}_{\mathrm{U}}{}^2$ may be either non-degenerate or first-order degenerate.*

   *(iii) Suppose $\mathcal{X}$ is a topological space, $k(x, \cdot)$ is continuous for each $x$, and $\sup_{x \in \mathcal{X}} k(x, x)$ is finite. Further assume the supports of $P$ and $Q$ are not disjoint. Then $\widehat{\mathrm{MMD}}_{\mathrm{U}}{}^2$ is degenerate if and only if $\mu_P = \mu_Q$.*

   *(iv) In Setting B, $\widehat{\mathrm{MMD}}_{\mathrm{U}}{}^2$ is degenerate if and only if $\mu_P = \mu_Q$.*

   *(v) In Setting B, $\langle \mu_P - \mu_Q, C_P(\mu_P - \mu_Q)\rangle > 0$ if and only if $\langle \mu_P - \mu_Q, C_Q(\mu_P - \mu_Q)\rangle > 0$.*

*Proof.* Because $C_P$ and $C_Q$ are each positive semi-definite, $C_P + C_Q = 0$ if and only if $C_P = 0 = C_Q$. Proposition 2.6 then immediately gives both the result about infinite degeneracy and (i).

**Proof of (ii)** Most usual settings are non-degenerate; for an explicit example, consider $\mathcal{X} = \mathbb{R}$, $P = \mathcal{N}(0,1)$, $Q = \mathcal{N}(1,1)$, and $k(x,y) = xy$. Then $\mu_P = (t \mapsto 0) = \mathbf{0}$, $\mu_Q = (t \mapsto t)$, $C_P = C_Q = \mathrm{Id}$, and so $\langle \mu_P - \mu_Q, (C_P + C_Q)(\mu_P - \mu_Q)\rangle = 2\|\mu_Q\|_{\mathcal{H}}^2 = 2\|k(\cdot, 1)\|_{\mathcal{H}}^2 = 2k(1,1) = 2$.

For a degenerate case, consider $\mathcal{X} = \mathbb{R}$, $k(x,y) = \max(1 - |x - y|, 0)$, $P = \mathrm{Uniform}(\{1,2\})$, and $Q = \mathrm{Uniform}(\{3,4\})$. These have nonzero kernel covariance operators: for instance,

$$
\langle k(1, \cdot), C_P k(1, \cdot)\rangle = \operatorname*{Var}_{X \sim P}[k(1, X)] = \tfrac{1}{2} \cdot 1^2 + \tfrac{1}{2} \cdot 0^2 - \left(\tfrac{1}{2} \cdot 1 + \tfrac{1}{2} \cdot 0\right)^2 = \tfrac{1}{2} - \tfrac{1}{4} = \tfrac{1}{4} > 0.
$$

We also have $\mu_P \neq \mu_Q$, as can be seen by considering their inner products with e.g. $k(2.2, \cdot)$, or because $P \neq Q$ and $k$ is characteristic (Sriperumbudur et al., 2008, Corollary 8). Notice, however, that $\Pr_{X \sim P, Y \sim Q}(k(X, Y) = 0) = 1$. Thus, if we write

$$\widehat{\mathrm{MMD}}_{\mathrm{U}}^2 = \frac{1}{n(n-1)} \sum_{i \neq j} k(x_i, x_j) + \frac{1}{n(n-1)} \sum_{i \neq j} k(y_i, y_j) - \frac{2}{n^2} \sum_{i \neq j} k(x_i, y_j),$$

the last sum is identically zero, leaving us with the sum of two independent U-statistics with kernel $k$. Applying Proposition 2.5 for the first such sum, $\mathrm{Var}[\mathbb{E}[k(X, X') \mid X]] = \mathrm{Var}_{X' \sim P}[\mu_P(X')] = 0$ because $\mu_P(1) = \mu_P(2) = \frac{1}{2}$, and $P = \mathrm{Uniform}(\{1, 2\})$; however, for any $x \in \{1, 2\}$, $\mathrm{Var}[k(x, X)] = \frac{1}{4}$ as seen above, and hence the variance is $\frac{2}{n(n-1)} \cdot \frac{1}{4} = \frac{1}{2n(n-1)}$. The other term is independent and has the same variance. Thus the variance of $\widehat{\mathrm{MMD}}_{\mathrm{U}}^2$ is $1/(n(n-1))$, so it must be first-order degenerate: we have a situation where $\mu_P \neq \mu_Q$, $C_P \neq 0$, $C_Q \neq 0$, and yet $\widehat{\mathrm{MMD}}_{\mathrm{U}}^2$ is degenerate.

**Proof of (iii) and (iv)**  If $\mu_P = \mu_Q$, then Proposition 2.6 immediately implies that $\widehat{\mathrm{MMD}}_{\mathrm{U}}^2$ is degenerate. Thus, suppose that $\widehat{\mathrm{MMD}}_{\mathrm{U}}^2$ is degenerate for $C_P$, $C_Q$ not both zero; we will show that $\mu_P = \mu_Q$, or equivalently that $\Delta = \mu_P - \mu_Q$ is the zero function. Since $C_P$ and $C_Q$ are positive semi-definite and we have assumed degeneracy, Proposition 2.6 implies that we must have

$$\langle \mu_P - \mu_Q, C_P(\mu_P - \mu_Q) \rangle = \mathrm{Var}_{X \sim P}(\Delta(X)) = 0 \quad \text{and} \quad \langle \mu_P - \mu_Q, C_Q(\mu_P - \mu_Q) \rangle = \mathrm{Var}_{Y \sim Q}(\Delta(Y)) = 0.$$

Thus, there must be $c_P, c_Q \in \mathbb{R}$ such that

$$\Pr_{X \sim P}(\Delta(X) = c_P) = 1 \qquad \text{and} \qquad \Pr_{Y \sim Q}(\Delta(Y) = c_Q) = 1.$$

But notice that

$$\|\mu_P - \mu_Q\|^2 = \langle \mu_P, \Delta \rangle - \langle \mu_Q, \Delta \rangle = \mathbb{E}_{X \sim P}[\Delta(X)] - \mathbb{E}_{Y \sim Q}[\Delta(Y)] = c_P - c_Q,$$

and so if $c_P = c_Q$ then $\mu_P = \mu_Q$.

For (iii), let $x$ be in the support of $P$, and suppose that $\Delta = \mu_P - \mu_Q$ has $\Delta(x) \neq c_P$. But then by continuity there is an open neighbourhood $N_x \ni x$ for which $|\Delta(x') - c_P| > \epsilon$ for all $x' \in N_x$; since $x$ is in the support of $P$, $P(N_x) > 0$, contradicting that $\Delta(x) = c_P$ $P$-almost surely. We similarly have $\Delta(y) = c_Q$ for all $y$ in the support of $Q$; thus, if there is a point in both supports, we must have $c_P = c_Q$.

For (iv), $k$ being bounded and real-analytic implies that each $f \in \mathcal{H}$ is as well (Chwialkowski et al., 2015, Lemma 1). As before, degeneracy implies $\Delta(x) = c_P$ for all $x$ in the support of $P$, and $\Delta(y) = c_Q$ for all $y$ in the support of $Q$. Mityagin (2015) shows that if a real-analytic function is zero on a set of positive Lebesgue measure, the function is identically zero; since $\Delta - c_P$ is analytic, it must be zero on all of $\mathcal{X}$. But since $\mathbb{R}^d$ is Hausdorff, the support of $Q$ is nonempty, and hence $c_Q = c_P$.

**Proof of (v):**  Reusing our previous notation, positive semi-definiteness implies $\langle \Delta, C_P \Delta \rangle, \langle \Delta, C_Q \Delta \rangle$ are either zero or positive. Suppose $\langle \Delta, C_P \Delta \rangle = 0$; then following the proof of (iv) we know that the function $\Delta$ is almost surely a constant $c_P$ on the support of $P$, and since it is also real-analytic and $P$'s support has positive Lebesgue measure, it is equal to $c_P$ everywhere. Thus $\langle \Delta, C_Q \Delta \rangle = \mathrm{Var}_{Y \sim Q} \Delta(Y) = 0$. The same reasoning applies for $Q$, and so $\langle \Delta, C_Q \Delta \rangle = 0$ implies $\langle \Delta, C_P \Delta \rangle = 0$. □

# D  Two Sample MMD Variance

We now decompose the variance of $\widehat{\mathrm{MMD}}^2$ based on the formula (7).

**Theorem 3.4.** *Under Setting A, it holds for any $n_X, n_Y \geq 2$ that*

$$
\operatorname*{Var}_{\substack{\mathbf{S}_P \sim P^{n_X} \\ \mathbf{S}_Q \sim Q^{n_Y}}} (\widehat{\mathrm{MMD}}^2(\mathbf{S}_P, \mathbf{S}_Q)) = \frac{4}{n_X}\left(1 - \frac{4}{n_Y} \cdot \frac{n_X - 2}{n_X - 1} \cdot \frac{n_Y - 2}{n_Y - 1}\right) \langle \mu_P - \mu_Q, C_P(\mu_P - \mu_Q) \rangle
$$

$$
+ \frac{4}{n_Y}\left(1 - \frac{4}{n_X} \cdot \frac{n_X - 2}{n_X - 1} \cdot \frac{n_Y - 2}{n_Y - 1}\right) \langle \mu_P - \mu_Q, C_Q(\mu_P - \mu_Q) \rangle
$$

$$
+ \frac{2}{n_X(n_X - 1)} \|C_P\|_{\mathrm{HS}}^2 + \frac{2}{n_Y(n_Y - 1)} \|C_Q\|_{\mathrm{HS}}^2 + \frac{4}{n_X n_Y} \langle C_P, C_Q \rangle_{\mathrm{HS}}
$$

$$
+ \frac{16}{n_X n_Y}\left(\frac{n_X - 2}{n_X - 1} \cdot \frac{n_Y - 2}{n_Y - 1}\right)\left[\langle \mu_P, C_P \mu_P \rangle + \langle \mu_Q, C_Q \mu_Q \rangle\right].
$$

*Proof.* We will need to compute various conditional expectations of the MMD estimator kernel function $h(x, x'; y, y') = \langle k(x, \cdot) - k(y, \cdot), k(x', \cdot) - k(y', \cdot) \rangle$. For brevity, in this proof we will use $\phi(x)$ to denote the feature map $k(x, \cdot)$. We can omit some calculations because of the additional symmetry $h(x, x'; y, y') = h(y, y'; x, x')$.

As $C_P, C_Q$ are self-adjoint, we have the useful identity

$$
\langle \mu_P, C_P \mu_P \rangle + \langle \mu_Q, C_P \mu_Q \rangle - 2\langle \mu_Q, C_P \mu_P \rangle = \left[\langle \mu_P, C_P \mu_P \rangle - \langle \mu_P, C_P \mu_Q \rangle\right] + \left[\langle \mu_Q, C_P \mu_Q \rangle - \langle \mu_Q, C_P \mu_P \rangle\right]
$$

$$
= \langle \mu_P, C_P(\mu_P - \mu_Q) \rangle + \langle \mu_Q, C_P(\mu_Q - \mu_P) \rangle
$$

$$
= \langle \mu_P - \mu_Q, C_P(\mu_P - \mu_Q) \rangle,
$$

and similarly $\langle \mu_Q, C_Q \mu_Q \rangle + \langle \mu_P, C_Q \mu_P \rangle - 2\langle \mu_P, C_Q \mu_Q \rangle = \langle \mu_P - \mu_Q, C_Q(\mu_P - \mu_Q) \rangle$.

**Order 1 terms.**

$$
\mathbb{E}(h(X, X'; Y, Y') \mid X) = \mu_P(X) + \mathbb{E}\, k(Y, Y') - \mu_Q(X) - \mathbb{E}\, k(X', Y)
$$

$$
\zeta_X = \operatorname{Var}(\mu_P(X) - \mu_Q(X) + \mathrm{const})
$$

$$
= \langle \mu_P - \mu_Q, C_P(\mu_P - \mu_Q) \rangle
$$

**Order 2 terms.**

$$
\mathbb{E}(h(X, X'; Y, Y')|X, X') = k(X, X') + \mathbb{E}\, k(Y, Y') - \mu_Q(X) - \mu_Q(X')
$$

$$
\operatorname{Var}(k(X, X')) = \mathbb{E}[k(X, X')^2] - (\mathbb{E}[k(X, X')])^2
$$

$$
= \mathbb{E}\langle \phi(X) \otimes \phi(X), \phi(X') \otimes \phi(X') \rangle_{\mathrm{HS}} - \|\mu_P\|^4
$$

$$
= (\|C_P\|_{\mathrm{HS}}^2 + 2\langle C_P, \mu_P \otimes \mu_P \rangle_{\mathrm{HS}} + \|\mu_P\|^4) - \|\mu_P\|^4
$$

$$
= \|C_P\|_{\mathrm{HS}}^2 + 2\langle \mu_P, C_P \mu_P \rangle
$$

$$
\operatorname{Cov}(\mu_Q(X), k(X, X')) = \mathbb{E}[\mu_Q(X) k(X, X')] - \mathbb{E}[k(X, Y)]\, \mathbb{E}[k(X, X')]
$$

$$
= \langle \mu_Q, C_P \mu_P \rangle
$$

$$
\zeta_{XX'} = \operatorname{Var}(k(X, X')) + \operatorname{Var}(\mu_Q(X)) + \operatorname{Var}(\mu_Q(X'))
$$

$$
- 2\operatorname{Cov}(\mu_Q(X), k(X, X')) - 2\operatorname{Cov}(\mu_Q(X'), k(X, X'))
$$

$$
= \|C_P\|_{\mathrm{HS}}^2 + 2\langle \mu_P, C_P \mu_P \rangle + 2\langle \mu_Q, C_P \mu_Q \rangle - 4\langle \mu_Q, C_P \mu_P \rangle
$$

$$
= \|C_P\|_{\mathrm{HS}}^2 + 2\langle \mu_P - \mu_Q, C_P(\mu_P - \mu_Q) \rangle
$$

Let $\tilde{C}_P = \mathbb{E}[\phi(X) \otimes \phi(X)]$ and define $\tilde{C}_Q$ analogously.

$$\mathbb{E}(h(X, X'; Y, Y')|X, Y) = \mu_P(X) + \mu_Q(Y) - \frac{1}{2}\left[\mu_Q(X) + \mu_P(Y) + k(X, Y) + \mathbb{E}\,k(X, Y)\right]$$

$$\begin{aligned}
\langle C_P, C_Q \rangle_{\mathrm{HS}} &= \langle \tilde{C}_P, \tilde{C}_Q \rangle_{\mathrm{HS}} - \langle \tilde{C}_P, \mu_Q \otimes \mu_Q \rangle_{\mathrm{HS}} - \langle \tilde{C}_Q, \mu_P \otimes \mu_P \rangle_{\mathrm{HS}} + \langle \mu_P, \mu_Q \rangle^2 \\
&= \mathbb{E}[k(X, Y)^2] - \langle C_P, \mu_Q \otimes \mu_Q \rangle_{\mathrm{HS}} - \langle C_Q, \mu_P \otimes \mu_P \rangle_{\mathrm{HS}} - \langle \mu_P, \mu_Q \rangle^2 \\
&= \mathrm{Var}(k(X, Y)) - \langle \mu_Q, C_P \mu_Q \rangle - \langle \mu_P, C_Q \mu_P \rangle
\end{aligned}$$

$$\begin{aligned}
\zeta_{XY} &= \mathrm{Var}\left(\left(\mu_P(X) - \frac{1}{2}\mu_Q(X)\right)\right) + \mathrm{Var}\left(\mu_Q(Y) - \frac{1}{2}\mu_P(Y)\right) + \frac{1}{4}\mathrm{Var}(k(X, Y)) \\
&\quad + \mathrm{Cov}\left(\mu_P(X) - \frac{1}{2}\mu_Q(X), k(X, Y)\right) + \mathrm{Cov}\left(\mu_Q(Y) - \frac{1}{2}\mu_P(Y), k(X, Y)\right) \\
&= \langle \mu_P - \frac{1}{2}\mu_Q, C_P(\mu_P - \frac{1}{2}\mu_Q) \rangle + \langle \mu_Q - \frac{1}{2}\mu_P, C_Q(\mu_Q - \frac{1}{2}\mu_P) \rangle \\
&\quad + \frac{1}{4}\langle C_P, C_Q \rangle_{\mathrm{HS}} + \frac{1}{4}\langle \mu_Q, C_P \mu_Q \rangle + \frac{1}{4}\langle \mu_P, C_Q \mu_P \rangle + \langle \mu_P - \frac{1}{2}\mu_Q, C_P \mu_Q \rangle \\
&\quad + \langle \mu_Q - \frac{1}{2}\mu_P, C_Q \mu_P \rangle \\
&= \frac{1}{4}\langle C_P, C_Q \rangle_{\mathrm{HS}} + \langle \mu_P, C_P \mu_P \rangle + \langle \mu_Q, C_Q \mu_Q \rangle,
\end{aligned}$$

where the last line follows by

$$\begin{aligned}
&\langle \mu_P - \frac{1}{2}\mu_Q, C_P(\mu_P - \frac{1}{2}\mu_Q) \rangle + \frac{1}{4}\langle \mu_Q, C_P \mu_Q \rangle + \langle \mu_P - \frac{1}{2}\mu_Q, C_P \mu_Q \rangle \\
&= \left[\langle \mu_P - \frac{1}{2}\mu_Q, C_P(\mu_P - \frac{1}{2}\mu_Q) \rangle + \langle \mu_P - \frac{1}{2}\mu_Q, C_P \frac{1}{2}\mu_Q \rangle\right] + \left[\frac{1}{4}\langle \mu_Q, C_P \mu_Q \rangle + \langle \frac{1}{2}\mu_P - \frac{1}{4}\mu_Q, C_P \mu_Q \rangle\right] \\
&= \langle \mu_P - \frac{1}{2}\mu_Q, C_P \mu_P \rangle + \langle \frac{1}{2}\mu_P, C_P \mu_Q \rangle = \langle \mu_P - \frac{1}{2}\mu_Q, C_P \mu_P \rangle + \langle \frac{1}{2}\mu_Q, C_P \mu_P \rangle = \langle \mu_P, C_P \mu_P \rangle.
\end{aligned}$$

**Order 3 terms.**

$$\mathbb{E}(h(X, X'; Y, Y') \mid X, X', Y) = k(X, X') + \mu_Q(Y) - \frac{1}{2}[k(X, Y) + \mu_Q(X') + k(X', Y) + \mu_Q(X)]$$

Let $R := k(X, X') - \frac{1}{2}[k(X, Y) + k(X', Y)]$, then we may write

$$\begin{aligned}
\zeta_{XX'Y} &= \mathrm{Var}(\mu_Q(Y)) + \frac{1}{4}\mathrm{Var}(\mu_Q(X')) + \frac{1}{4}\mathrm{Var}(\mu_Q(X)) + \mathrm{Var}(R) \\
&\quad + 2\,\mathrm{Cov}(\mu_Q(Y), R) - \mathrm{Cov}(\mu_Q(X'), R) - \mathrm{Cov}(\mu_Q(X), R) \\
&= \langle \mu_Q, C_Q \mu_Q \rangle + \frac{1}{2}\langle \mu_Q, C_P \mu_Q \rangle + \mathrm{Var}(R) + 2\,\mathrm{Cov}(\mu_Q(Y), R) \\
&\quad - 2\,\mathrm{Cov}(\mu_Q(X'), R).
\end{aligned}$$

We may simplify the terms including $R$

$$
\begin{aligned}
\mathrm{Var}(R) &= \mathrm{Var}(k(X,X')) + \frac{1}{4}\mathrm{Var}(k(X,Y)) + \frac{1}{4}\mathrm{Var}(k(X',Y)) - \mathrm{Cov}(k(X,X'),k(X,Y)) \\
&\quad - \mathrm{Cov}(k(X,X'),k(X',Y)) + \frac{1}{2}\mathrm{Cov}(k(X,Y),k(X',Y)) \\
&= \|C_P\|_{\mathrm{HS}}^2 + 2\langle \mu_P, C_P\mu_P\rangle + \frac{1}{2}\left[\langle C_P, C_Q\rangle_{\mathrm{HS}} + \langle \mu_Q, C_P\mu_Q\rangle + \langle \mu_P, C_Q\mu_P\rangle\right] \\
&\quad - 2\langle \mu_P, C_P\mu_Q\rangle + \frac{1}{2}\langle \mu_P, C_Q\mu_P\rangle \\
&= \|C_P\|_{\mathrm{HS}}^2 + 2\langle \mu_P, C_P\mu_P\rangle + \frac{1}{2}\langle C_P, C_Q\rangle_{\mathrm{HS}} + \frac{1}{2}\langle \mu_Q, C_P\mu_Q\rangle + \langle \mu_P, C_Q\mu_P\rangle - 2\langle \mu_P, C_P\mu_Q\rangle
\end{aligned}
$$

$$
\mathrm{Cov}(\mu_Q(Y),R) = \mathrm{Cov}\left(\mu_Q(Y), -\frac{1}{2}[k(X,Y)+k(X',Y)]\right) = -\mathrm{Cov}(\mu_Q(Y),k(X,Y)) = -\langle \mu_Q, C_Q\mu_P\rangle
$$

$$
\mathrm{Cov}(\mu_Q(X'),R) = \mathrm{Cov}\left(\mu_Q(X'), k(X,X') - \frac{1}{2}k(X',Y)\right) = \langle \mu_Q, C_P\mu_P\rangle - \frac{1}{2}\langle \mu_Q, C_P\mu_Q\rangle
$$

Plugging the above into the variance equation gives

$$
\begin{aligned}
\zeta_{XX'Y} &= \langle \mu_Q, C_Q\mu_Q\rangle + \frac{1}{2}\langle \mu_Q, C_P\mu_Q\rangle + \|C_P\|_{\mathrm{HS}}^2 + 2\langle \mu_P, C_P\mu_P\rangle \\
&\quad + \frac{1}{2}\langle C_P, C_Q\rangle_{\mathrm{HS}} + \frac{1}{2}\langle \mu_Q, C_P\mu_Q\rangle + \langle \mu_P, C_Q\mu_P\rangle - 2\langle \mu_P, C_P\mu_Q\rangle \\
&\quad - 2\langle \mu_Q, C_Q\mu_P\rangle - 2\langle \mu_Q, C_P\mu_P\rangle + \langle \mu_Q, C_P\mu_Q\rangle \\
&= \|C_P\|_{\mathrm{HS}}^2 + \frac{1}{2}\langle C_P, C_Q\rangle_{\mathrm{HS}} + 2[\langle \mu_P, C_P\mu_P\rangle + \langle \mu_Q, C_P\mu_Q\rangle - 2\langle \mu_P, C_P\mu_Q\rangle] \\
&\quad + [\langle \mu_Q, C_Q\mu_Q\rangle + \langle \mu_P, C_Q\mu_P\rangle - 2\langle \mu_Q, C_Q\mu_P\rangle] \\
&= \|C_P\|_{\mathrm{HS}}^2 + \frac{1}{2}\langle C_P, C_Q\rangle_{\mathrm{HS}} + 2\langle \mu_P - \mu_Q, C_P(\mu_P - \mu_Q)\rangle \\
&\quad + \langle \mu_P - \mu_Q, C_Q(\mu_P - \mu_Q)\rangle \\
&= \|C_P\|_{\mathrm{HS}}^2 + \frac{1}{2}\langle C_P, C_Q\rangle_{\mathrm{HS}} + \langle \mu_P - \mu_Q, (2C_P + C_Q)(\mu_P - \mu_Q)\rangle
\end{aligned}
$$

**Order 4 terms.** Let $R' := k(Y,Y') - \frac{1}{2}[k(X',Y') + k(X,Y')]$, we have $h(X,X';Y,Y') = R + R'$. Similar to the calculation for $R$ we have

$$
\mathrm{Var}(R') = \|C_Q\|_{\mathrm{HS}}^2 + 2\langle \mu_Q, C_Q\mu_Q\rangle + \frac{1}{2}\langle C_P, C_Q\rangle_{\mathrm{HS}} + \frac{1}{2}\langle \mu_Q, C_P\mu_Q\rangle + \langle \mu_P, C_Q\mu_P\rangle - 2\langle \mu_Q, C_Q\mu_P\rangle.
$$

Next we have

$$
\begin{aligned}
2\mathrm{Cov}(R,R') &= 2\mathrm{Cov}\left(k(X,X') - \frac{1}{2}[k(X,Y)+k(X',Y)], k(Y,Y') - \frac{1}{2}[k(X',Y')+k(X,Y')]\right) \\
&= -\mathrm{Cov}(k(X,X'),k(X',Y')+k(X,Y')) - \mathrm{Cov}(k(Y,Y'),k(X,Y)+k(X',Y)) \\
&\quad + \frac{1}{2}\mathrm{Cov}(k(X,Y)+k(X',Y),k(X',Y')+k(X,Y')) \\
&= -2\langle \mu_P, C_P\mu_Q\rangle - 2\langle \mu_Q, C_Q\mu_P\rangle + \langle \mu_Q, C_P\mu_Q\rangle.
\end{aligned}
$$

Finally we have

$$
\begin{aligned}
\zeta_{XX'YY'} &= \mathrm{Var}(R) + \mathrm{Var}(R') + 2\mathrm{Cov}(R,R') \\
&= \|C_P\|_{\mathrm{HS}}^2 + \|C_Q\|_{\mathrm{HS}}^2 + \langle C_P, C_Q\rangle_{\mathrm{HS}} + 2[\langle \mu_P, C_P\mu_P\rangle + \langle \mu_Q, C_P\mu_Q\rangle - 2\langle \mu_P, C_P\mu_Q\rangle] \\
&\quad + 2[\langle \mu_Q, C_Q\mu_Q\rangle + \langle \mu_P, C_Q\mu_P\rangle - 2\langle \mu_Q, C_Q\mu_P\rangle] \\
&= \|C_P\|_{\mathrm{HS}}^2 + \|C_Q\|_{\mathrm{HS}}^2 + \langle C_P, C_Q\rangle_{\mathrm{HS}} + 2\langle \mu_P - \mu_Q, (C_P + C_Q)(\mu_P - \mu_Q)\rangle
\end{aligned}
$$

**Simplification** Plugging the $\zeta$ values into (7) yields the variance for $\widehat{\mathrm{MMD}}^2$ of

$$
\frac{4}{n_X}\left(\frac{n_X-2}{n_X-1}\cdot\frac{n_Y-2}{n_Y}\cdot\frac{n_Y-3}{n_Y-1}\right)\zeta_X + \frac{4}{n_Y}\left(\frac{n_X-2}{n_X}\cdot\frac{n_X-3}{n_X-1}\cdot\frac{n_Y-2}{n_Y-1}\right)\zeta_Y
$$

$$
+\frac{2}{n_X(n_X-1)}\left(\frac{n_Y-2}{n_Y}\cdot\frac{n_Y-3}{n_Y-1}\right)\zeta_{XX'} + \frac{2}{n_Y(n_Y-1)}\left(\frac{n_X-2}{n_X}\cdot\frac{n_X-3}{n_X-1}\right)\zeta_{YY'}
$$

$$
+\frac{16}{n_Xn_Y}\left(\frac{n_X-2}{n_X-1}\cdot\frac{n_Y-2}{n_Y-1}\right)\zeta_{XY}
$$

$$
+\frac{8}{n_X(n_X-1)n_Y}\left(\frac{n_Y-2}{n_Y-1}\right)\zeta_{XX'Y} + \frac{8}{n_Xn_Y(n_Y-1)}\left(\frac{n_X-2}{n_X-1}\right)\zeta_{XYY'}
$$

$$
+\frac{4}{n_X(n_X-1)n_Y(n_Y-1)}\zeta_{XX'YY'},
$$

where

$$
\zeta_X = \langle\mu_P-\mu_Q, C_P(\mu_P-\mu_Q)\rangle \qquad \zeta_Y = \langle\mu_P-\mu_Q, C_Q(\mu_P-\mu_Q)\rangle
$$

$$
\zeta_{XX'} = \|C_P\|_{\mathrm{HS}}^2 + 2\langle\mu_P-\mu_Q, C_P(\mu_P-\mu_Q)\rangle \qquad \zeta_{YY'} = \|C_Q\|_{\mathrm{HS}}^2 + 2\langle\mu_P-\mu_Q, C_Q(\mu_P-\mu_Q)\rangle
$$

$$
\zeta_{XY} = \frac{1}{4}\langle C_P, C_Q\rangle_{\mathrm{HS}} + \langle\mu_P, C_P\mu_P\rangle + \langle\mu_Q, C_Q\mu_Q\rangle
$$

$$
\zeta_{XX'Y} = \|C_P\|_{\mathrm{HS}}^2 + \frac{1}{2}\langle C_P, C_Q\rangle_{\mathrm{HS}} + \langle\mu_P-\mu_Q, (2C_P+C_Q)(\mu_P-\mu_Q)\rangle
$$

$$
\zeta_{XYY'} = \|C_Q\|_{\mathrm{HS}}^2 + \frac{1}{2}\langle C_P, C_Q\rangle_{\mathrm{HS}} + \langle\mu_P-\mu_Q, (C_P+2C_Q)(\mu_P-\mu_Q)\rangle
$$

$$
\zeta_{XX'YY'} = \|C_P\|_{\mathrm{HS}}^2 + \|C_Q\|_{\mathrm{HS}}^2 + \langle C_P, C_Q\rangle_{\mathrm{HS}} + 2\langle\mu_P-\mu_Q, (C_P+C_Q)(\mu_P-\mu_Q)\rangle.
$$

Let $\nu_P = \langle\mu_P-\mu_Q, C_P(\mu_P-\mu_Q)\rangle$ and $\nu_Q = \langle\mu_P-\mu_Q, C_Q(\mu_P-\mu_Q)\rangle$. Then

$$
\zeta_X = \nu_P \qquad \zeta_Y = \nu_Q
$$

$$
\zeta_{XX'} = \|C_P\|_{\mathrm{HS}}^2 + 2\nu_P \qquad \zeta_{YY'} = \|C_Q\|_{\mathrm{HS}}^2 + 2\nu_Q
$$

$$
\zeta_{XY} = \frac{1}{4}\langle C_P, C_Q\rangle_{\mathrm{HS}} + \langle\mu_P, C_P\mu_P\rangle + \langle\mu_Q, C_Q\mu_Q\rangle
$$

$$
\zeta_{XX'Y} = \|C_P\|_{\mathrm{HS}}^2 + \frac{1}{2}\langle C_P, C_Q\rangle_{\mathrm{HS}} + 2\nu_P + \nu_Q
$$

$$
\zeta_{XYY'} = \|C_Q\|_{\mathrm{HS}}^2 + \frac{1}{2}\langle C_P, C_Q\rangle_{\mathrm{HS}} + \nu_P + 2\nu_Q
$$

$$
\zeta_{XX'YY'} = \|C_P\|_{\mathrm{HS}}^2 + \|C_Q\|_{\mathrm{HS}}^2 + \langle C_P, C_Q\rangle_{\mathrm{HS}} + 2\nu_P + 2\nu_Q.
$$

This means that $\mathrm{Var}(\widehat{\mathrm{MMD}}^2)$ is

$$
\frac{4}{n_X}\left(\frac{n_X-2}{n_X-1}\cdot\frac{n_Y-2}{n_Y}\cdot\frac{n_Y-3}{n_Y-1}\right)\nu_P + \frac{4}{n_Y}\left(\frac{n_X-2}{n_X}\cdot\frac{n_X-3}{n_X-1}\cdot\frac{n_Y-2}{n_Y-1}\right)\nu_Q
$$

$$
+\frac{2}{n_X(n_X-1)}\left(\frac{n_Y-2}{n_Y}\cdot\frac{n_Y-3}{n_Y-1}\right)(\|C_P\|_{\mathrm{HS}}^2+2\nu_P) + \frac{2}{n_Y(n_Y-1)}\left(\frac{n_X-2}{n_X}\cdot\frac{n_X-3}{n_X-1}\right)(\|C_Q\|_{\mathrm{HS}}^2+2\nu_Q)
$$

$$
+\frac{4}{n_Xn_Y}\left(\frac{n_X-2}{n_X-1}\cdot\frac{n_Y-2}{n_Y-1}\right)[\langle C_P, C_Q\rangle_{\mathrm{HS}}+4\langle\mu_P, C_P\mu_P\rangle+4\langle\mu_Q, C_Q\mu_Q\rangle]
$$

$$
+\frac{8}{n_X(n_X-1)n_Y}\left(\frac{n_Y-2}{n_Y-1}\right)\left[\|C_P\|_{\mathrm{HS}}^2+\frac{1}{2}\langle C_P, C_Q\rangle_{\mathrm{HS}}+2\nu_P+\nu_Q\right]
$$

$$
+\frac{8}{n_Xn_Y(n_Y-1)}\left(\frac{n_X-2}{n_X-1}\right)\left[\|C_Q\|_{\mathrm{HS}}^2+\frac{1}{2}\langle C_P, C_Q\rangle_{\mathrm{HS}}+\nu_P+2\nu_Q\right]
$$

$$
+\frac{4}{n_X(n_X-1)n_Y(n_Y-1)}\left[\|C_P\|_{\mathrm{HS}}^2+\|C_Q\|_{\mathrm{HS}}^2+\langle C_P, C_Q\rangle_{\mathrm{HS}}+2\nu_P+2\nu_Q\right].
$$

Gathering $\nu_P$ terms, we get a coefficient of $4/n_X$ times

$$\frac{(n_X - 2)(n_Y - 2)(n_Y - 3) + (n_Y - 2)(n_Y - 3) + 4(n_Y - 2) + 2(n_X - 2) + 2}{(n_X - 1)n_Y(n_Y - 1)}$$

$$= \frac{(n_X - 1)(n_Y - 2)(n_Y - 3) + 4(n_Y - 2) + 2(n_X - 1)}{(n_X - 1)n_Y(n_Y - 1)}$$

$$= \frac{(n_Y - 2)(n_Y - 3) + 2}{n_Y(n_Y - 1)} + \frac{4(n_Y - 2)}{(n_X - 1)n_Y(n_Y - 1)}$$

$$= \frac{n_Y^2 - 5n_Y + 8}{n_Y(n_Y - 1)} + \frac{4(n_Y - 2)}{(n_X - 1)n_Y(n_Y - 1)}$$

$$= \frac{n_Y(n_Y - 1) - 4(n_Y - 2)}{n_Y(n_Y - 1)} + \frac{4(n_Y - 2)}{(n_X - 1)n_Y(n_Y - 1)}$$

$$= 1 - \frac{4(n_Y - 2)}{n_Y(n_Y - 1)}\left(1 - \frac{1}{n_X - 1}\right)$$

$$= 1 - \frac{4}{n_Y} \cdot \frac{n_X - 2}{n_X - 1} \cdot \frac{n_Y - 2}{n_Y - 1};$$

the $\nu_Q$ terms are symmetric, so that term's coefficient is

$$\frac{4}{n_Y}\left(1 - \frac{4}{n_X} \cdot \frac{n_X - 2}{n_X - 1} \cdot \frac{n_Y - 2}{n_Y - 1}\right).$$

Next gathering the $\|C_P\|_{\mathrm{HS}}^2$ terms, we find a coefficient of

$$\frac{2}{n_X(n_X - 1)} \cdot \frac{(n_Y - 2)(n_Y - 3) + 4(n_Y - 2) + 2}{n_Y(n_Y - 1)} = \frac{2}{n_X(n_X - 1)} \frac{n_Y^2 - n_Y}{n_Y(n_Y - 1)} = \frac{2}{n_X(n_X - 1)},$$

and the $\|C_Q\|_{\mathrm{HS}}^2$ terms are again symmetric for a coefficient of $2/(n_Y(n_Y - 1))$. Finally, the $\langle C_P, C_Q \rangle_{\mathrm{HS}}$ terms gather to a coefficient of $4/(n_X n_Y)$, since

$$\frac{(n_X - 2)(n_Y - 2) + (n_Y - 2) + (n_X - 2) + 1}{(n_X - 1)(n_Y - 1)} = \frac{(n_X - 1)(n_Y - 2) + (n_X - 1)}{(n_X - 1)(n_Y - 1)} = \frac{(n_X - 1)(n_Y - 1)}{(n_X - 1)(n_Y - 1)} = 1.$$

The result follows. $\qquad\square$

## E  Asymptotics of Generalized U-statistics

This section characterizes the asymptotic behaviour of generalized U-statistics by filling in and generalizing the approach of Serfling (1980, Chapter 5). Throughout the section we fix a generalized U-statistic $U_n$ constructed from the kernel $h(x_{11}, \ldots, x_{m_1 1}, \ldots, x_{1c}, \ldots, x_{m_c c})$ and random samples $\mathbf{S} = \{X_{ij} : j \in [c], i \in [n_j]\}$, where $(X_{ij})_{i \in [n_j]} \sim \mu_j^{n_j}$ for each $j$. For brevity we write $n = n_{\min} = \min\{n_1, \ldots, n_c\}$. Additionally define

$$\mathbf{X} = (X_{ij} : j \in [c], \ i \in [m_j]) \sim \prod_{j=1}^{c} \mu_j^{m_j}, \quad \theta = \mathbb{E}[h(\mathbf{X})], \quad \tilde{h} = h - \theta.$$

To make our presentation more concise, we introduce an abbreviation to our expression in Definition 3.1 with $U_n = \prod_{j=1}^{c} \binom{n_j}{m_j}^{-1} \sum_{\mathbb{X}} h(\mathbb{X})$, $\mathbb{X}$ varying over each distinct collection of arguments in $\mathbf{S}$. The following definition generalizes the projection in Serfling (1980, section 5.3.4) to the case of generalized U-statistics, which itself is a simplification of Hoeffding's decomposition in his seminal work on the almost sure behaviour of one-sample U-statistics (Hoeffding, 1961, lemma 1). Janson and Nowicki (1991) consider a special case of these projections for two-sample U-statistics where one sample represents graph vertices and the other graph edges.

**Definition E.1.** *The order $r$ projection of $U_n$ is given by*

$$\hat{U}_{n,r} = \theta + \sum_{(i_1 j_1), \ldots, (i_r j_r)} \left( \mathbb{E}[U_n \mid X_{i_1 j_1}, \ldots, X_{i_r j_r}] - \theta \right),$$

*where the sum is over each subset of* **S** *of size $r$.*

Often we will simplify the projection of $U_n$ using its order of degeneracy. This is made precise by the following proposition.

**Proposition E.2.** *Let $\mathbb{X}, \mathbb{X}'$ be subsets of the sample* **S** *where $\mathbb{X} \sim \prod_{j=1}^{c} \mu_j^{m_j}$, then $\mathbb{E}[h(\mathbb{X}) \mid \mathbb{X}'] = \mathbb{E}[h(\mathbb{X}) \mid \mathbb{X} \cap \mathbb{X}']$ where we interpret $\mathbb{E}[h(\mathbb{X}) \mid \emptyset] = \mathbb{E}[h(\mathbb{X})]$. In particular if $U_n$ is order $r$ degenerate and $|\mathbb{X} \cap \mathbb{X}'| \leq r$ then $\mathbb{E}[\tilde{h}(\mathbb{X}) \mid \mathbb{X}'] = 0$.*

*Proof.* Since **S** consists of independent variables, $\mathbb{E}[h(\mathbb{X}) \mid \mathbb{X}']$ is obtained by integrating out the arguments in $h(\mathbb{X})$ which are not shared with $\mathbb{X}'$. It then follows that $\mathbb{E}[h(\mathbb{X}) \mid \mathbb{X}'] = \mathbb{E}[h(\mathbb{X}) \mid \mathbb{X} \cap \mathbb{X}']$. Now if $U_n$ is order $r$ degenerate and $|\mathbb{X} \cap \mathbb{X}'| \leq r$, the symmetries of $h$ imply $\mathrm{Var}(\mathbb{E}[\tilde{h}(\mathbb{X})|\mathbb{X} \cap \mathbb{X}']) = 0$ and in turn $\mathbb{E}[\tilde{h}(\mathbb{X})|\mathbb{X}'] = \mathbb{E}[\tilde{h}(\mathbb{X})] = 0$. $\square$

The following lemma shows that if $U_n$ has order-$r$ degeneracy (identifying $r = 0$ with non-degeneracy) then its asymptotic distribution can be approximated by $\hat{U}_{n,r+1}$ in the mean-square norm.

**Lemma E.3.** *If $U_n$ is a generalized U-statistic with order of degeneracy at least $r$, then $\mathbb{E}\,|U_n - \hat{U}_{n,r+1}|^2 = \mathcal{O}(n^{-(r+2)})$.*

*Proof.* Expanding $\mathbb{E}[U_n \mid X_{i_1 j_1}, \ldots, X_{i_{r+1} j_{r+1}}]$ yields

$$U_n - \hat{U}_{n,r+1} = \prod_{j=1}^{c} \binom{n_j}{m_j}^{-1} \sum_{\mathbb{X}} \left[ \tilde{h}(\mathbb{X}) - \sum_{(i_1 j_1), \ldots, (i_{r+1} j_{r+1})} \mathbb{E}[\tilde{h}(\mathbb{X}) \mid X_{i_1 j_1}, \ldots, X_{i_{r+1} j_{r+1}}] \right].$$

Turning to the inner sum, Proposition E.2 implies $\mathbb{E}[\tilde{h}(\mathbb{X}) \mid X_{i_1 j_1}, \ldots, X_{i_{r+1} j_{r+1}}] = 0$ if $\mathbb{X}$ and $X_{i_1 j_1}, \ldots, X_{i_{r+1} j_{r+1}}$ share less than $r + 1$ terms. Thus removing these zero-valued terms leaves us with

$$U_n - \hat{U}_{n,r} = \prod_{j=1}^{c} \binom{n_j}{m_j}^{-1} \sum_{\mathbb{X}} \left[ \tilde{h}(\mathbb{X}) - \sum_{(i_1 j_1), \ldots, (i_{r+1} j_{r+1}) \in \mathbb{X}} \mathbb{E}[\tilde{h}(\mathbb{X}) \mid X_{i_1 j_1}, \ldots, X_{i_{r+1} j_{r+1}}] \right], \quad (15)$$

where $(i_1 j_1), \ldots, (i_{r+1} j_{r+1}) \in \mathbb{X}$ denotes that $\mathbb{X}$ contains $X_{i_1 j_1}, \ldots, X_{r+1, j_{r+1}}$. The above puts $U_n - \hat{U}_{n,r}$ in the form a generalized U-statistic and furthermore conditioning on any $r + 1$ terms in **X** gives

$$\mathbb{E}\left[ \tilde{h}(\mathbf{X}) - \sum_{(i_1 j_1), \ldots, (i_{r+1} j_{r+1}) \in \mathbf{X}} \mathbb{E}[\tilde{h}(\mathbf{X}) \mid X_{i_1 j_1}, \ldots, X_{i_{r+1} j_{r+1}}] \,\middle|\, X_{s_1 t_1}, \ldots, X_{s_{r+1} t_{r+1}} \right]$$

$$= \mathbb{E}[\tilde{h}(\mathbf{X}) \mid X_{s_1 t_1}, \ldots, X_{s_{r+1} t_{r+1}}] - \mathbb{E}[\tilde{h}(\mathbf{X}) \mid X_{s_1 t_1}, \ldots, X_{s_{r+1} t_{r+1}}] = 0 \quad \text{by Proposition E.2.}$$

If we had conditioned on fewer than $r + 1$ variables, the term above would be zero again due to $U_n$'s order of degeneracy. Therefore $U_n - \hat{U}_{n,r}$'s order of degeneracy is at least $r + 1$. Since $U_n - \hat{U}_{n,r}$ has zero mean, it follows from the variance formula (8) that $\mathbb{E}\,|U_n - \hat{U}_{n,r}|^2 = \mathcal{O}(n^{-(r+2)})$. $\square$

The next two results builds on our lemma and deduces the asymptotic distribution of generalized U-statistics based on their first and second order projections.

**Theorem 3.8.** *Let $U_{n_{\min}}$ be a c-sample U-statistic with kernel $h$, where $n_{\min} = \min\{n_1, \ldots, n_c\}$. If $n_{\min}/n_j \to \rho_j \in [0, 1]$ for each $j \in [c]$, then*

$$\sqrt{n_{\min}}(U_{n_{\min}} - \mathbb{E}[h(\mathbf{X})]) \xrightarrow{d} \mathcal{N}\left(0, \sum_{j=1}^{c} \rho_j m_j^2 \operatorname{Var}\left(\mathbb{E}\left[h(\mathbf{X}) \mid X_{1j}\right]\right)\right),$$

*where $\mathcal{N}(0, 0)$ is interpreted as a point mass at 0. In particular, if $U_{n_{\min}}$ is non-degenerate and each $\rho_j > 0$ (the proportional regime), the distribution above is normal with positive variance.*

*Proof.* By Lemma E.3 we have that

$$\mathbb{E}[|\sqrt{n}(U_n - \hat{U}_{n,1})|^2] = n \, \mathbb{E}[|U_n - \hat{U}_{n,1}|^2] = n \, \mathcal{O}(n^{-2}) = \mathcal{O}(n^{-1}),$$

so it suffices to deduce the distribution of $\sqrt{n}(\hat{U}_{n,1} - \theta)$. Define $\tilde{h}_j = \mathbb{E}[\tilde{h}(\mathbf{X}) \mid X_{1j} = \cdot]$, we may write

$$\hat{U}_{n,1} - \theta = \prod_{j=1}^{c} \binom{n_j}{m_j}^{-1} \sum_{j=1}^{c} \sum_{i=1}^{n_j} \sum_{\mathbb{X}} \mathbb{E}[\tilde{h}(\mathbb{X}) \mid X_{ij}]$$

Note that if $X_{ij}$ is not contained in $\mathbb{X}$ then Proposition E.2 implies $\mathbb{E}[\tilde{h}(\mathbb{X}) \mid X_{ij}] = 0$, thus counting the number of $\mathbb{X}$ containing $X_{ij}$ leaves us with

$$\hat{U}_{n,1} - \theta = \prod_{j=1}^{c} \binom{n_j}{m_j}^{-1} \sum_{j=1}^{c} \sum_{i=1}^{n_j} \binom{n_j - 1}{m_j - 1} \prod_{j' \neq j} \binom{n_{j'}}{m_{j'}} \tilde{h}_j(X_{ij}) = \sum_{j=1}^{c} \frac{m_j}{n_j} \sum_{i=1}^{n_j} \tilde{h}_j(X_{ij}).$$

If $\operatorname{Var}(\mathbb{E}[h(\mathbf{X}) \mid X_{1j}]) > 0$, we may combine the central limit theorem and Slutsky's theorem to get

$$\sqrt{n}\left(\frac{m_j}{n_j} \sum_{i=1}^{n_j} \tilde{h}_j(X_{ij})\right) = \sqrt{\frac{n}{n_j}} m_j \left(\frac{1}{\sqrt{n_j}} \sum_{i=1}^{n_j} \tilde{h}_j(X_{ij})\right) \xrightarrow{d} \mathcal{N}\left(0, \rho_j m_j^2 \operatorname{Var}(\mathbb{E}[h(\mathbf{X}) \mid X_{1j}])\right).$$

Otherwise if $\operatorname{Var}(\mathbb{E}[h(\mathbf{X}) \mid X_{1j}]) = 0$ then $\sum_{i=1}^{n_j} \tilde{h}_j(X_{ij})$ is almost surely zero. The desired limiting distribution follows by applying the continuous mapping theorem to $\sqrt{n}(\hat{U}_{n,1} - \theta)$. If $U_n$ is non-degenerate then $\operatorname{Var}(\mathbb{E}[h(\mathbf{X}) \mid X_{1j}]) > 0$ for some $j$, thus proportionality guarantees that the variance is positive. $\qquad\square$

To deal with the asymmetric kernel associated with second order projections of generalized U-statistics, we have a preliminary result analogous to a singular value decomposition (SVD) for square-integrable kernels which is stated in Koroljuk and Borovskich (1993, equation 4.5.21) without proof.

**Lemma E.4** (Kernel SVD). *Let $k \in L_2(\mathcal{X} \times \mathcal{Y}, \mu \times \nu)$ and $L_2(\mathcal{X}, \mu), L_2(\mathcal{Y}, \nu)$ be separable. Then $k$ has the representation as an $L_2$ limit*

$$k(x, y) = \sum_n \sigma_n v_n(x) u_n(y),$$

*where $\sigma_n$ are the singular values of the operator $T : L_2(\mathcal{X}, \mu) \to L_2(\mathcal{Y}, \nu)$ defined by*

$$T : f \mapsto \int k(x, \cdot) f(x) \, d\mu(x),$$

*and $\{v_n\} \subset L_2(\mathcal{X}, \mu), \{u_n\} \subset L_2(\mathcal{Y}, \nu)$ are orthonormal.*

*Proof.* Since $k$ is square integrable, $T$ is Hilbert-Schmidt and hence compact. Letting $T^*$ denote the adjoint, $T^*T$ is a compact self-adjoint operator and hence provides a countable orthonormal basis of eigenvectors

$\{v_n\}$ of $T^*T$ with corresponding eigenvectors $\{\lambda_n\}$ arranged so that $\lambda_1 \geq \lambda_2 \geq \cdots \geq 0$. For all non-zero $\lambda_n$, put $\sigma_n = \sqrt{\lambda_n}$ and $u_n = \sigma_n^{-1}Tv_n$. Notice that

$$\langle u_n, u_m \rangle = \frac{1}{\sigma_n\sigma_m}\langle Tv_n, Tv_m \rangle = \frac{1}{\sigma_n\sigma_m}\langle T^*Tv_n, v_m \rangle = \frac{\sigma_n}{\sigma_m}\langle v_n, v_m \rangle$$

so $\{u_n\}$ is orthonormal in $L_2(\mathcal{Y}, \nu)$. Thus we have

$$\iint \left| k(x,y) - \sum_{n=1}^{L} \sigma_n v_n(x)u_n(y) \right|^2 d\mu(x)d\nu(y)$$

$$= \iint \left( |k(x,y)|^2 - 2\sum_{n=1}^{L} \sigma_n k(x,y)v_n(x)u_n(y) + \sum_{n=1}^{L} \sigma_n^2 v_n^2(x)u_n^2(y) \right) d\mu(x)d\nu(y)$$

$$= \|k\|_2^2 - 2\sum_{n=1}^{L} \sigma_n \int u_n(y)(Tv_n)(y) \, d\nu(y) + \sum_{n=1}^{L} \sigma_n^2$$

$$= \|k\|_2^2 - \sum_{n=1}^{L} \sigma_n^2 = \|k\|_2^2 - \sum_{n=1}^{L}\langle T^*Tv_n, v_n\rangle_2 = \|k\|_2^2 - \sum_{n=1}^{L}\|Tv_n\|_2^2.$$

It is a standard result that $\|k\|_2 = \|T\|_{\mathrm{HS}}$. Therefore the final term vanishes as $L \to \infty$ . $\qquad\square$

Observe that $\hat{U}_{n,2}$ is a linear combination of the conditional kernels

$$\tilde{h}_{jj} := \mathbb{E}(\tilde{h}(\mathbf{X}) \mid X_{1j} = \cdot, X_{2j} = \cdot), \quad 1 \leq j \leq c$$
$$\tilde{h}_{st} := \mathbb{E}(\tilde{h}(\mathbf{X}) \mid X_{1s} = \cdot, X_{1t} = \cdot), \quad 1 \leq s < t \leq c.$$

Now assuming each $\tilde{h}_{jj} \in L_2(\mu_j^2)$ and $\tilde{h}_{st} \in L_2(\mu_s \times \mu_t)$ leads to a variety of kernel decompositions. For the symmetric kernel $\tilde{h}_{jj}$, Dunford and Schwartz (1988, Section VI, Exercises 44 and 56) give the decomposition

$$\tilde{h}_{jj}(x_1, x_2) = \sum_{l=1}^{\infty} \lambda_{jl}\psi_{jl}(x_1)\psi_{jl}(x_2), \tag{16}$$

where the limit is taken with in $L_2$ norm, and $\{\psi_{jl}\}, \{\lambda_{jl}\}$ are the (orthonormal) eigenfunction-eigenvalue pairs of $T_j := f \mapsto \mathbb{E}_{X\sim\mu_j}[\tilde{h}_{jj}(\cdot, X)f(X)]$. The aforementioned Lemma E.4 applies to $\tilde{h}_{st}$ giving

$$\tilde{h}_{st}(x_1, x_2) = \sum_{l=1}^{\infty} \sigma_{stl}v_{stl}(x_1)u_{stl}(x_2). \tag{17}$$

These results are applicable to the coming proof.

**Theorem E.5.** *Let $U_n$ be a c-sample generalized U-statistic with an associated kernel $h$, where $n = \min\{n_1, \ldots, n_c\}$ and $\rho_j := \lim_{n\to\infty} n/n_j$ exists for each $j$. Further assume $\tilde{h}_{jj} \in L_2(\mu_j^2)$ for $j \in [c]$ and $\tilde{h}_{st} \in L_2(\mu_s \times \mu_t)$ for $1 \leq s < t \leq c$, admitting the decompositions (16) and (17). If $U_n$ is first-order degenerate then $n(U_n - \theta)$ converges in distribution to $Y$, where*

$$Y = \sum_{j=1}^{c} \binom{m_j}{2}\rho_j \sum_{l=1}^{\infty} \lambda_{jl}(a_{jl}^2 - 1) + \sum_{s<t} m_s m_t \sqrt{\rho_s\rho_t} \sum_{l=1}^{\infty} \sigma_{stl}b_{stl}c_{stl},$$

*and $\{a_{jl}\}, \{b_{stl}\}, \{c_{stl}\}$ are marginally $\mathcal{N}(0,1)$ variables obtained from the weak limits*

$$\frac{1}{\sqrt{n_j}}\sum_{i=1}^{n_j} \psi_{jl}(X_{ij}) \to_d a_{jl}, \quad \frac{1}{\sqrt{n_s}}\sum_{i=1}^{n_s} v_{stl}(X_{is}) \to_d b_{stl}, \quad \frac{1}{\sqrt{n_t}}\sum_{i=1}^{n_t} u_{stl}(X_{it}) \to_d c_{stl}.$$

*Proof.* By Lemma E.3, $n(U_n - \theta)$ converges in $L_2$ to $V_n := n(\hat{U}_{n,2} - \theta)$. Writing the projection explicitly

$$n^{-1}V_n = \sum_{j=1}^{c} \sum_{i_1 < i_2} [\mathbb{E}(U_n \mid X_{i_1 j}, X_{i_2 j}) - \theta] + \sum_{s<t} \sum_{i_1, i_2} [\mathbb{E}(U_n \mid X_{i_1 s}, X_{i_2 t}) - \theta],$$

which may be simplified by removing the zero-valued terms according to Proposition E.2

$$\sum_{i_1 < i_2} [\mathbb{E}(U_n \mid X_{i_1 j}, X_{i_2 j}) - \theta] = \prod_{j'=1}^{c} \binom{n_{j'}}{m_{j'}}^{-1} \sum_{i_1 < i_2} \sum_{\mathbb{X}} \mathbb{E}(\tilde{h}(\mathbb{X}) \mid X_{i_1 j}, X_{i_2 j})$$

$$= \prod_{j'=1}^{c} \binom{n_{j'}}{m_{j'}}^{-1} \sum_{i_1 < i_2} \binom{n_j - 2}{m_j - 2} \prod_{j' \neq j} \binom{n_{j'}}{m_{j'}} \tilde{h}_{jj}(X_{i_1 j}, X_{i_2 j})$$

$$= \binom{m_j}{2}\binom{n_j}{2}^{-1} \sum_{i_1 < i_2} \tilde{h}_{jj}(X_{i_1 j}, X_{i_2 j})$$

$$\sum_{i_1, i_2} [\mathbb{E}(U_n \mid X_{i_1 s}, X_{i_2 t}) - \theta] = \prod_{j=1}^{c} \binom{n_j}{m_j}^{-1} \sum_{i_1, i_2} \sum_{\mathbb{X}} \mathbb{E}(\tilde{h}(\mathbb{X}) \mid X_{i_1 s}, X_{i_2 t})$$

$$= \prod_{j=1}^{c} \binom{n_j}{m_j}^{-1} \sum_{i_1, i_2} \binom{n_s - 1}{m_s - 1}\binom{n_t - 1}{m_t - 1} \prod_{j \neq s, t} \binom{n_j}{m_j} \tilde{h}_{st}(X_{i_2, s}, X_{i_2 t})$$

$$= \frac{m_s m_t}{n_s n_t} \sum_{i_1, i_2} \tilde{h}_{st}(X_{i_2, s}, X_{i_2 t}).$$

Thus we're left with

$$n^{-1}V_n = \sum_{j=1}^{c} \binom{m_j}{2}\binom{n_j}{2}^{-1} \sum_{i_1 < i_2} \tilde{h}_{jj}(X_{i_1 j}, X_{i_2 j}) + \sum_{s<t} \frac{m_s m_t}{n_s n_t} \sum_{i_1, i_2} \tilde{h}_{st}(X_{i_1 s}, X_{i_2 t}).$$

Replacing $\tilde{h}_{jj}$ and $\tilde{h}_{st}$ with their $L_2$ expansions, we define the truncation of $V_n$ to $L$ eigenfunctions

$$n^{-1}V_{nL} := \sum_{j=1}^{c} \binom{m_j}{2}\binom{n_j}{2}^{-1} \sum_{l=1}^{L} \sum_{i_1 < i_2} \lambda_{jl} \psi_{jl}(X_{i_1 j}) \psi_{jl}(X_{i_2 j})$$

$$+ \sum_{s<t} \frac{m_s m_t}{n_s n_t} \sum_{l=1}^{L} \sum_{i_1, i_2} \sigma_{stl} v_{stl}(X_{i_1 s}) u_{stl}(X_{i_2 t}). \tag{18}$$

and the truncation of $Y$ by

$$Y_L := \sum_{j=1}^{c} \binom{m_j}{2} \rho_j \sum_{l=1}^{L} \lambda_{jl}(a_{jl}^2 - 1) + \sum_{s<t} m_s m_t \sqrt{\rho_s \rho_t} \sum_{l=1}^{L} \sigma_{stl} b_{stl} c_{stl}.$$

Using $V_{nL}$ and $Y_L$ as points of comparison, we have the following error decomposition in terms of characteristic functions:

$$|\mathbb{E}\exp(ixV_n) - \mathbb{E}\exp(ixY)| \leq |\mathbb{E}\exp(ixV_n) - \mathbb{E}\exp(ixV_{nL})| + |\mathbb{E}\exp(ixV_{nL}) - \mathbb{E}\exp(ixY_L)|$$
$$+ |\mathbb{E}\exp(ixY_L) - \mathbb{E}\exp(ixY)|.$$

The rest of the proof is divided in three sections, each focused on bounding a term above.

**First term:** We first apply the inequality $|\exp(iz) - 1| \leq |z|$

$$|x|^{-1} \mathbb{E} |\exp(ixV_n) - \exp(ixV_{nL})|$$

$$\leq \mathbb{E}|V_n - V_{nL}| \leq [\mathbb{E}|V_n - V_{nL}|^2]^{1/2} \qquad \text{by Hölder's inequality}$$

$$\leq \sum_{j=1}^{c} \binom{m_j}{2} \underbrace{\left[ \mathbb{E} \left| n \binom{n_j}{2}^{-1} \sum_{l=L+1}^{\infty} \sum_{i_1 < i_2} \lambda_{jl} \psi_{jl}(X_{i_1 j}) \psi_{jl}(X_{i_2 j}) \right|^2 \right]^{1/2}}_{A_j}$$

$$+ \sum_{s<t} m_s m_t \underbrace{\left[ \mathbb{E} \left| \frac{n}{n_s n_t} \sum_{l=L+1}^{\infty} \sum_{i_1, i_2} \sigma_{stl} v_{stl}(X_{i_1 s}) u_{stl}(X_{i_2 t}) \right|^2 \right]^{1/2}}_{B_{st}}.$$

Since $V_{nL} \to V_n$ in $L_2$ we have

$$A_j^2 = n^2 \binom{n_j}{2}^{-2} \mathbb{E} \left| \sum_{l=L+1}^{\infty} \sum_{i_1 < i_2} \lambda_{jl} \psi_{jl}(X_{i_1 j}) \psi_{jl}(X_{i_2 j}) \right|^2$$

$$= n^2 \binom{n_j}{2}^{-2} \sum_{l=L+1}^{\infty} \lambda_{jl}^2 \mathbb{E} \left| \sum_{i_1 < i_2} \psi_{jl}(X_{i_1 j}) \psi_{jl}(X_{i_2 j}) \right|^2 \qquad \text{by orthogonality}$$

$$= n^2 \binom{n_j}{2}^{-2} \sum_{l=L+1}^{\infty} \lambda_{jl}^2 \sum_{i_1 < i_2} \sum_{i_3 < i_4} \mathbb{E}[\psi_{jl}(X_{i_1 j}) \psi_{jl}(X_{i_2 j}) \psi_{jl}(X_{i_3 j}) \psi_{jl}(X_{i_4 j})].$$

First-order degeneracy implies

$$\mathbb{E}[\tilde{h}_{jj}(\cdot, X_{ij})] = \sum_{l=1}^{\infty} \lambda_{jl} \mathbb{E}[\psi_{jl}(X_{ij})] \psi_{jl}(\cdot) = 0$$

so by linear independence of the eigenfunctions we must have $\mathbb{E}[\psi_{jl}(X_{ij})] = 0$ if $\lambda_{jl} \neq 0$. It follows by independence that

$$\mathbb{E}[\psi_{jl}(X_{i_1 j}) \psi_{jl}(X_{i_2 j}) \psi_{jl}(X_{i_3 j}) \psi_{jl}(X_{i_4 j})] = \begin{cases} 1 & \text{if } (i_1, i_2) = (i_3, i_4) \\ 0 & \text{otherwise.} \end{cases}$$

Hence we get

$$A_j^2 = n^2 \binom{n_j}{2}^{-1} \sum_{l=L+1}^{\infty} \lambda_{jl}^2 = \frac{2n^2}{n_j(n_j - 1)} \sum_{l=L+1}^{\infty} \lambda_{jl}^2 \leq 4 \sum_{l=L+1}^{\infty} \lambda_{jl}^2.$$

We may similarly show that $\mathbb{E}[v_{stl}(X_{i_1 s})] = \mathbb{E}[u_{stl}(X_{i_2 t})] = 0$ if $\sigma_{stl} \neq 0$, thus

$$B_{st}^2 = \left( \frac{n}{n_s n_t} \right)^2 \mathbb{E} \left| \sum_{l=L+1}^{\infty} \sum_{i_1, i_2} \sigma_{stl} v_{stl}(X_{i_1 s}) u_{stl}(X_{i_2 t}) \right|^2$$

$$= \left( \frac{n}{n_s n_t} \right)^2 \sum_{l=L+1}^{\infty} \sigma_{stl}^2 \mathbb{E} \left| \sum_{i_1, i_2} v_{stl}(X_{i_1 s}) u_{stl}(X_{i_2 t}) \right|^2 = \frac{n^2}{n_s n_t} \sum_{l=L+1}^{\infty} \sigma_{stl}^2 \leq \sum_{l=L+1}^{\infty} \sigma_{stl}^2.$$

Since the eigenvalues $\{\lambda_{jl}\}$ and singular values $\{\sigma_{stl}\}$ belong to Hilbert-Schmidt operators, their squared series converge. Therefore $\lim_{L \to \infty} \mathbb{E}|\exp(ixV_n) - \exp(ixV_{nL})| = 0$ as it is a linear combination of the $A_j, B_{st}$. Importantly, our choice of $L$ is independent of $n$.

**Second term:** Fix any $L \geq 1$, we will focus on the following terms in expression (18) of $V_{nL}$

$$\alpha_j := \frac{2}{n_j} \sum_{l=1}^{L} \sum_{i_1 < i_2} \lambda_{jl} \psi_{jl}(X_{i_1 j}) \psi_{jl}(X_{i_2 j})$$

$$\beta_{st} := \frac{1}{\sqrt{n_s n_t}} \sum_{l=1}^{L} \sum_{i_1, i_2} \sigma_{stl} v_{stl}(X_{i_1 s}) u_{stl}(X_{i_2 t}),$$

which we may rewrite as

$$\alpha_j = \frac{1}{n_j} \sum_{l=1}^{L} \lambda_{jl} \left( \left( \sum_{i=1}^{n_j} \psi_{jl}(X_{ij}) \right)^2 - \sum_{i=1}^{n_j} \psi_{jl}^2(X_{ij}) \right)$$

$$= \sum_{l=1}^{L} \lambda_{jl} \left( \left( \frac{1}{\sqrt{n_j}} \sum_{i=1}^{n_j} \psi_{jl}(X_{ij}) \right)^2 - \frac{1}{n_j} \sum_{i=1}^{n_j} \psi_{jl}^2(X_{ij}) \right)$$

$$\beta_{st} = \sum_{l=1}^{L} \sigma_{stl} \left( \frac{1}{\sqrt{n_s}} \sum_{i_1=1}^{n_s} v_{stl}(X_{i_1 s}) \right) \left( \frac{1}{\sqrt{n_t}} \sum_{i_2=1}^{n_t} u_{stl}(X_{i_2 t}) \right).$$

Note that $\frac{1}{n_j} \sum_{i=1}^{n_j} \psi_{jl}^2(X_{ij}) \to 1$ a.s. by the law of large numbers. By the above we may view $V_{nL}$ as a continuous function in terms of the sample means and $(Z_1, \ldots, Z_c)$, where $Z_j$ is the vector concatenating

$$\left( \frac{1}{\sqrt{n_j}} \sum_{i=1}^{n_j} \psi_{jl}(X_{ij}) : 1 \leq l \leq L \right)^{\top}$$

$$\left( \frac{1}{\sqrt{n_j}} \sum_{i=1}^{n_j} v_{jtl}(X_{ij}) : 1 \leq l \leq L, \ j < t \right)^{\top}$$

$$\left( \frac{1}{\sqrt{n_j}} \sum_{i=1}^{n_j} u_{sjl}(X_{ij}) : 1 \leq l \leq L, \ s < j \right)^{\top}.$$

That is, $Z_j$ contains all normalized samples from $\mu_j$. The central limit theorem gives $Z_j \to_d \mathcal{N}(0, \Sigma_j)$ as $n_j \to \infty$, where $\Sigma_j$ is the covariance matrix between pairs of $\psi_{jl}(X_{ij}), v_{jtl}(X_{ij}), u_{sjl}(X_{ij})$. Moreover $Z_j$ is independent across each $j$, so we have the joint convergence $(Z_1, \ldots, Z_c) \to_d \mathcal{N}(0, \Sigma)$ as $n \to \infty$, where $\Sigma$ is the block diagonal matrix $\text{diag}(\Sigma_1, \ldots, \Sigma_c)$. A final application of the continuous mapping theorem and Slutsky's theorem gives

$$V_{nL} = \sum_{j=1}^{c} \binom{m_j}{2} \frac{n}{n_j - 1} \alpha_j + \sum_{s<t} m_s m_t \frac{n}{\sqrt{n_s n_t}} \beta_{st}$$

$$\longrightarrow_d \sum_{j=1}^{c} \binom{m_j}{2} \rho_j \sum_{l=1}^{L} \lambda_{jl}(a_{jl}^2 - 1) + \sum_{s<t} m_s m_t \sqrt{\rho_s \rho_t} \sum_{l=1}^{L} \sigma_{stl} b_{stl} c_{stl} = Y_L.$$

Convergence in distribution accordingly gives $|\mathbb{E} \exp(ix V_{nL}) - \mathbb{E} \exp(ix Y_L)| \to 0$ as $n \to \infty$.

**Third term:** Recall that $Y$ is the $L_2$ limit of $Y_L$, so again using $|\exp(iz) - 1| \leq |z|$ we get

$$|\mathbb{E} \exp(ix Y_L) - \mathbb{E} \exp(ix Y)| \leq |x| [\mathbb{E} |Y_L - Y|^2]^{1/2} \to 0 \text{ as } L \to \infty.$$

To complete the proof, pick $\varepsilon > 0$. For $L$ sufficiently large the first and third term are less than $\varepsilon/3$. Fixing this $L$, we may pick $N$ such that the second term is bounded by $\varepsilon/3$ for all $n \geq N$, giving

$$|\mathbb{E} \exp(ix V_n) - \mathbb{E} \exp(ix Y)| < \varepsilon, \quad \forall n \geq N. \qquad \square$$

We can now determine the null distribution of $\widehat{\mathrm{MMD}}^2$ as a specialized case of the preceding theorem. We shall see this distribution has a much cleaner form than above due to the additional symmetries of the MMD kernel.

**Theorem 3.7.** *Assume Setting A and that* $\mathrm{MMD}^2(P,Q) = 0$. *Assume* $\min\{n_X, n_Y\}/n_X \to \rho_X$ *and* $\min\{n_X, n_Y\}/n_Y \to \rho_Y$ *for some* $\rho_X, \rho_Y$ *in* $[0,1]$. $\widehat{\mathrm{MMD}}^2$ *converges in distribution as*

$$\min\{n_X, n_Y\} \widehat{\mathrm{MMD}}^2 \xrightarrow{d} (\rho_X + \rho_Y) \sum_{l=1}^{\infty} \lambda_l \left( Z_l^2 - 1 \right),$$

*where each* $Z_l$ *is independently* $\mathcal{N}(0,1)$, *and the* $\lambda_l$ *are the eigenvalues of the integral equation*

$$\mathbb{E}_X \left[ \langle \phi(X) - \mu_P, \phi(y) - \mu_P \rangle g(X) \right] = \lambda g(y). \tag{9}$$

*Proof.* Theorem 3.4 implies that $\widehat{\mathrm{MMD}}^2$ is first-order degenerate, so we will apply Theorem E.5. Define

$$\tilde{h}_{XX} = \mathbb{E}[h(X, X'; Y, Y') \mid X = \cdot, X' = \cdot]$$
$$\tilde{h}_{YY} = \mathbb{E}[h(X, X'; Y, Y') \mid Y = \cdot, Y' = \cdot]$$
$$\tilde{h}_{XY} = \mathbb{E}[h(X, X'; Y, Y') \mid X = \cdot, Y = \cdot]$$

Since $\mathrm{MMD}^2(P,Q) = 0$ implies $\mu_P = \mu_Q$, we calculate the conditional kernels as

$$\begin{aligned}
\tilde{h}_{XX}(X, X') &= k(X, X') + \mathbb{E}\, k(Y, Y') - \mu_Q(X) - \mu_Q(X') \\
&= \langle \phi(X) - \mu_Q, \phi(X) - \mu_Q \rangle \\
&= \langle \phi(X) - \mu_P, \phi(X) - \mu_P \rangle \\
\tilde{h}_{XY}(X, Y) &= \mu_P(X) + \mu_Q(Y) - \frac{1}{2} \left( \mu_Q(X) + \mu_P(Y) + k(X, Y) + \mathbb{E}\, k(X', Y') \right) \\
&= \frac{1}{2} \left( \mu_P(X) + \mu_P(Y) - k(X, Y) - \mathbb{E}\, k(X', Y') \right) \\
&= -\frac{1}{2} \langle \phi(X) - \mu_P, \phi(Y) - \mu_P \rangle
\end{aligned}$$

and by symmetry $\tilde{h}_{YY}(Y, Y') = \langle \phi(X) - \mu_P, \phi(Y) - \mu_P \rangle$ as well. It follows that the integral operators of $\tilde{h}_{XX}, \tilde{h}_{YY}, \tilde{h}_{XY}$ share the same normalized eigenfunctions, and the eigenvalues of $\tilde{h}_{XY}$ carry an extra factor of $-1/2$. Now Proposition 2.1 implies $\tilde{h}_{XX}, \tilde{h}_{YY}, \tilde{h}_{XY}$ are square integrable with respect to $P^2, Q^2, P \times Q$, so following Theorem E.5

$$\min\{n_x, n_y\} \widehat{\mathrm{MMD}}^2 \to_d \rho_X \sum_{l=1}^{\infty} \lambda_l(a_l^2 - 1) + \rho_Y \sum_{l=1}^{\infty} \lambda_l(b_l^2 - 1) - 2\sqrt{\rho_X \rho_Y} \sum_{l=1}^{\infty} \lambda_l a_l b_l,$$

where $a_l, b_l \sim \mathcal{N}(0,1)$ are independent. Finally we may complete the square, simplifying the above to

$$\sum_{l=1}^{\infty} \lambda_l \left[ (\rho_X^{1/2} a_l - \rho_Y^{1/2} b_l)^2 - (\rho_X + \rho_Y) \right] = \sum_{l=1}^{\infty} \lambda_l \left[ (\rho_X + \rho_Y) Z_l^2 - (\rho_X + \rho_Y) \right] = (\rho_X + \rho_Y) \sum_{l=1}^{\infty} \lambda_l(Z_l^2 - 1),$$

as desired. $\qquad\square$

# F Proof of uniform convergence

We will now show that the uniform convergence analysis of Liu et al. (2020), also holds for our estimators. Throughout this section, we use the notation of that paper: $k_\omega$ refers to a kernel with parameters $\omega \in \Omega$, for which $\nu := \sup_{\omega \in \Omega} \sup_{x \in \mathcal{X}} k_\omega(x, x)$ and $L_k = \sup_{\omega \neq \omega' \in \Omega} \sup_{x, y \in \mathcal{X}} |k_\omega(x, y) - k_{\omega'}(x, y)| / \|\omega - \omega'\|$.

**Theorem F.1.** *Theorem 11, as well as Corollaries 12 and 13, of Liu et al. (2020) all hold exactly as written for our* $\widehat{\mathrm{MMD}}^2$ *and* $\hat{\sigma}$ *when* $n$ *is replaced by* $n_{\mathrm{harm}} = \frac{2}{1/n_X + 1/n_Y}$.

*Proof.* The proof is identical to that of Liu et al. (2020) with their Propositions 15 and 16 replaced by Propositions F.2 and F.3. $\qquad\square$

The following two uniform convergence results follow the same strategy as the analogous results of Liu et al. (2020), which are their Propositions 15 and 16, but require minor changes.

**Proposition F.2.** *Proposition 15 of Liu et al. (2020) also holds for* $\widehat{\mathrm{MMD}}^2$ *when $n$ is replaced by $n_{\mathrm{harm}} = \frac{2}{1/n_X + 1/n_Y}$.*

*Proof.* First, note that the estimator $\widehat{\mathrm{MMD}}^2$ of (1) satisfies bounded differences. If we replace one entry of $\mathbf{S}_P$, the $\mathbf{S}_P$-to-$\mathbf{S}_P$ average changes $n_X - 1$ of its $\binom{n_X}{2}$ summands, each by at worst $2\nu$. Thus that term changes by at most $\frac{4\nu}{n_X}$. The $\mathbf{S}_Q$-to-$\mathbf{S}_Q$ average does not change, while the $\mathbf{S}_P$-to-$\mathbf{S}_Q$ average changes $n_Y$ of its $n_X n_Y$ terms again by at most $2\nu$. Since this term is doubled, the total change in $\widehat{\mathrm{MMD}}^2$ is at most $\frac{8\nu}{n_X}$. Similarly, replacing one entry of $\mathbf{S}_Q$ changes $\widehat{\mathrm{MMD}}^2$ by at most $\frac{8\nu}{n_Y}$. McDiarmid's inequality therefore tells us that the subgaussian parameter of $\widehat{\mathrm{MMD}}^2$ is at most $\frac{1}{2}\sqrt{n_X \cdot \frac{(8\nu)^2}{n_X^2} + n_Y \cdot \frac{(8\nu)^2}{n_X^2}} = 4\nu\sqrt{\frac{1}{n_X} + \frac{1}{n_Y}} = 4\nu\sqrt{2/n_{\mathrm{harm}}}$. As $\hat{\eta}$ is also unbiased, for any given parameter $\omega$, it holds with probability at least $1 - \delta$ that

$$|\hat{\eta}_\omega - \eta_\omega| \leq \frac{8\nu}{\sqrt{n_{\mathrm{harm}}}}\sqrt{\log\frac{2}{\delta}}.$$

The coefficient in the proof of Proposition 15 of Liu et al. (2020) was instead $8\sqrt{2}\nu/\sqrt{n}$; we can recover their bound on this component with $n_{\mathrm{harm}}$ instead of $n$ because $8 < 8\sqrt{2}$.

We also have that for any two $\omega, \omega' \in \Omega$, $|\hat{\eta}_\omega(\mathbf{S}_P, \mathbf{S}_Q) - \hat{\eta}_{\omega'}(\mathbf{S}_P, \mathbf{S}_Q)|$ is upper-bounded by the average of $|k_\omega(x, x') - k_{\omega'}(x, x')|$, plus that of $|k_\omega(y, y') - k_{\omega'}(y, y')|$, plus twice that of $|k_\omega(x, y) - k_{\omega'}(x, y)|$. Each of these terms is at most $L_k\|\omega - \omega'\|$, so this difference is at most $4L_k\|\omega - \omega'\|$. The same holds for $\eta_\omega - \eta_{\omega'}$, giving a Lipschitz constant for the error of at most $8L_k$. This is the same as that of Proposition 15 of Liu et al. (2020).

These are the two properties used to develop a standard covering number bound in the remainder of the original proof; given these properties, the proof is identical.[4] $\qquad\square$

**Proposition F.3.** *Proposition 16 of Liu et al. (2020) also holds for our $\hat{\sigma}^2$ when $n$ is replaced by $n_{\mathrm{harm}} = \frac{2}{1/n_X + 1/n_Y}$.*

*Proof.* The proof is identical once we replace their Lemmas 17 to 19 with our Lemmas F.4 to F.6. $\qquad\square$

**Lemma F.4.** *Lemma 17 of Liu et al. (2020) also holds for our $\hat{\sigma}^2$ when $n$ is replaced by $n_{\mathrm{harm}}$.*

*Proof.* For (10), let $A_j = \frac{1}{n_X}\sum_{i=1}^{n_X} k(x_i, x_j)$, so that the estimator is $\frac{1}{n_X}\sum_{i=1}^{n_X} A_j^2 - \left(\frac{1}{n_X}\sum_{i=1}^{n_X} A_j\right)^2$. Changing one element of $\mathbf{S}_P$, say $x_1$, changes $A_1$ by at most $2\nu$ and all other $A_j$ by at most $2\nu/n_X$. Since each $|A_j| \leq \nu$, we have that $|(A_j')^2 - A_j^2| = |A_j' + A_j||A_j' - A_j| \leq 2\nu|A_j' - A_j|$. Thus the change in the mean of $A_j^2$ is at most $\frac{2\nu}{n_X}\left(2\nu + (n_X - 1)\frac{2\nu}{n_X}\right) = 4\nu^2\left(\frac{2}{n_X} - \frac{1}{n_X^2}\right)$. The change in the mean of $A_j$ is at most $\frac{1}{n_X}2\nu + \frac{n_X - 1}{n_X}\frac{2\nu}{n_X} = 2\nu\left(\frac{2}{n_X} - \frac{1}{n_X^2}\right)$, and so the change in the square of the mean of $A_j$ is at most $2\nu$ times that. Adding these together, the change in the full estimator (10) is at most $\frac{16\nu^2}{n_X} - \frac{8\nu^2}{n_X^2}$.

For (11), let $B_i = \frac{1}{n_Y}\sum_{j=1}^{n_Y} k(x_i, y_j)$, so the overall estimator is $\frac{1}{n_X}\sum_{i=1}^{n_X} B_i^2 - \left(\frac{1}{n_X}\sum_{i=1}^{n_X} B_i\right)^2$. Changing a single element of $\mathbf{S}_P$, say $x_1$, changes $B_1$ by at most $2\nu$, and thus $B_1^2$ by at most $4\nu^2$; the other $B_i$ are unchanged. Thus the overall estimator changes by at most $\frac{4\nu^2}{n_X} + 2\nu\frac{2\nu}{n_X} = \frac{8\nu^2}{n_X}$. Changing a single element

---

[4]It is worth noting, but not carrying through in our analysis here, that the constant 4 used by Liu et al. from Cucker and Smale (2001, Proposition 5) could be replaced by 3 with no further assumption (e.g. Sutherland, 2025, Proposition 4.11).

of $\mathbf{S}_Q$ instead changes each $B_i$ by at most $2\nu/n_Y$, and hence each $B_i^2$ by at most $4\nu^2/n_Y$; this makes the change in the overall estimator at most $8\nu^2/n_Y$.

The estimator (12) is $\frac{1}{n_X}\sum_{i=1}^{n_X} A_i B_i - \left(\frac{1}{n_X}\sum_{i=1}^{n_X} A_i\right)\left(\frac{1}{n_X}\sum_{i=1}^{n_X} B_i\right)$. Changing $x_1$ changes $A_1$ by up to $2\nu$ and the other $A_i$ by up to $2\nu/n_X$, and changes $B_1$ by at most $2\nu$ while leaving the other $B_i$ unchanged. Using

$$|A_i' B_i' - A_i B_i| \le |A_i' B_i' - A_i' B_i| + |A_i' B_i - A_i B_i| = |A_i'||B_i' - B_i| + |B_i||A_i' - A_i|,$$

the change in $A_1 B_1$ is at most $\nu(2\nu+2\nu) = 4\nu^2$, while the change in the other $A_i B_i$ is at most $2\nu^2/n_X$. Thus the total change to $\frac{1}{n_X}\sum_{i=1}^{n_X} A_i B_i$ is at most $4\nu^2/n_X + \frac{n_X-1}{n_X}2\nu^2/n_X = 6\nu^2/n_X - 2\nu^2/n_X^2$. For the other term, the change to the mean of the $A_i$ is at most $4\nu/n_X - 2\nu/n_X^2$, and to the mean of the $B_i$ is at most $2\nu/n_X$. Thus the term changes by at most $\nu\left(6\nu/n_X - 2\nu/n_X^2\right)$, for a total change in (12) from changing $x_1$ of at most $12\nu^2/n_X - 4\nu^2/n_X^2$.

If we instead change $y_1$, then all of the $A_i$ are unchanged, while each $B_i$ changes by at most $2\nu/n_Y$. Thus each $A_i B_i$ changes by at most $2\nu^2/n_Y$, and so does its mean over $i$. The mean of the $A_i$ is unchanged, while the mean of the $B_i$ changes by at most $2\nu/n_Y$, giving a total change in (12) of at most $4\nu^2/n_Y$.

Combining these three pieces, our estimator $\hat{\zeta}_X$ changes by at most

$$\frac{16\nu^2}{n_X} - \frac{8\nu^2}{n_X^2} + \frac{8\nu^2}{n_X} + 2\cdot\frac{12\nu^2}{n_X} - 2\cdot\frac{4\nu^2}{n_X^2} = \frac{48\nu^2}{n_X} - \frac{16\nu^2}{n_X^2} \le \frac{48\nu^2}{n_X}$$

when changing a single element of $\mathbf{S}_P$, or by at most

$$0 + \frac{8\nu^2}{n_Y} + 2\cdot\frac{4\nu^2}{n_Y} = \frac{16\nu^2}{n_Y}$$

if we change a single element of $\mathbf{S}_Q$. The case for $\zeta_Y$ is symmetric, so changing a single element of $\mathbf{S}_P$ changes the estimator $\hat{\sigma}^2 = 4\rho_X\hat{\zeta}_X + 4\rho_Y\hat{\zeta}_Y$ by $4(48\rho_X + 16\rho_Y)\nu^2/n_X = 64(3\rho_X + \rho_Y)\nu^2/n_X$, while changing a single element of $\mathbf{S}_Q$ changes $\hat{\sigma}^2$ by at most $64(\rho_X + 3\rho_Y)\nu^2/n_Y$. Thus, the total subgaussian parameter of $\hat{\sigma}^2$ is at most $\frac{1}{2}\sqrt{n_X\cdot\frac{64^2(3\rho_X+\rho_Y)^2}{n_X^2} + n_Y\cdot\frac{64^2(\rho_X+3\rho_Y)^2}{n_Y^2}} \le \frac{1}{2}\cdot 64\cdot 4\sqrt{\frac{1}{n_X}+\frac{1}{n_Y}} = 128\sqrt{\frac{2}{n_{\mathrm{harm}}}}$, using that $\rho_X, \rho_Y \in [0,1]$. The equivalent parameter found in Lemma 17 of Liu et al. (2020) is $448 > 128\sqrt{2}$. $\qquad\square$

**Lemma F.5.** *Lemma 18 of Liu et al. (2020) holds for our $\hat{\sigma}^2$ when $n$ is replaced by $n_{\mathrm{harm}}$.*

*Proof.* Each of the estimators can be viewed as a sum of various V-statistics. In general, a V statistic has bias introduced only by the terms with repeated indices, and thus its bias is bounded by the portion of repeated indices. We can write (10) as

$$\frac{1}{n_X^3}\sum_{i=1}^{n_X}\sum_{j=1}^{n_X}\sum_{l=1}^{n_X} k(x_i,x_j)k(x_i,x_l) - \frac{1}{n_X^4}\sum_{i=1}^{n_X}\sum_{j=1}^{n_X}\sum_{a=1}^{n_X}\sum_{b=1}^{n_X} k(x_i,x_j)k(x_a,x_b),$$

where each individual term has absolute value at most $\nu^2$. The number of repeated indices in the first, third-order V-statistic is $n_X^3 - n_X(n_X-1)(n_X-2) = n_X^3 - n_X(n_X^2 - 3n_X + 2) = 3n_X^2 - 2n_X$, and its bias is less than $\nu^2\frac{3n_X^2}{n_X^3} = \frac{3\nu^2}{n_X}$. For the fourth-order term, there are $n_X^4 - n_X(n_X-1)(n_X-2)(n_X-3) = 6n_X^3 - 11n_X^2 + 6n_X$ potentially biased terms, for a total bias of less than $\frac{6\nu^2}{n_X}$. So the bias of (10) is less than $\frac{9\nu^2}{n_X}$.

For (11), the first term has $n_X n_Y$ repeat indices out of $n_X n_Y^2$ total terms, and the second has less than $n_X^2 n_Y + n_X n_Y^2$ out of $n_X^2 n_Y^2$, for a total bias at most $\nu^2\left(\frac{1}{n_X} + \frac{2}{n_Y}\right)$.

For (12), we have $n_X n_Y$ possible repeats out of $n_X^2 n_Y$ terms and less than $3n_X^2 n_Y$ out of $n_X^3 n_Y$ for the second part, giving a total bias at most $4\nu^2/n_X$.

Thus, $\hat{\zeta}_X$ has bias at most $\frac{9\nu^2}{n_X} + \frac{\nu^2}{n_X} + \frac{2\nu^2}{n_Y} + \frac{8\nu^2}{n_X} = \frac{18\nu^2}{n_X} + \frac{2\nu^2}{n_Y}$. Similarly, $\hat{\zeta}_Y$'s bias is at most $\frac{18\nu^2}{n_Y} + \frac{2\nu^2}{n_X}$, and so $\hat{\sigma}^2$ has bias at most

$$4\left(\frac{18\rho_X + 2\rho_Y}{n_X} + \frac{18\rho_Y + 2\rho_X}{n_Y}\right)\nu^2 \le \frac{160\nu^2}{n_{\mathrm{harm}}},$$

using that $\rho_X, \rho_Y \in [0,1]$ and $\frac{1}{n_X} + \frac{1}{n_Y} = \frac{2}{n_{\mathrm{harm}}}$. This is indeed smaller than Lemma 18 of Liu et al. (2020)'s bound of $1152\nu^2/n$, after replacing $n$ by $n_{\mathrm{harm}}$. $\qquad\square$

**Lemma F.6.** *Lemma 19 of Liu et al. (2020) also holds for our $\hat{\sigma}_\omega^2$ and $\sigma_\omega^2$.*

*Proof.* First notice that

$$
\begin{aligned}
|k_\omega(s,t)k_\omega(u,v) - k_{\omega'}(s,t)k_{\omega'}(u,v)| &\le |k_\omega(s,t)k_\omega(u,v) - k_\omega(s,t)k_{\omega'}(u,v)| + |k_\omega(s,t)k_{\omega'}(u,v) - k_{\omega'}(s,t)k_{\omega'}(u,v)| \\
&= |k_\omega(s,t)||k_\omega(u,v) - k_{\omega'}(u,v)| + |k_\omega(s,t) - k_{\omega'}(s,t)||k_{\omega'}(u,v)| \\
&\le 2\nu L_k \|\omega - \omega'\|.
\end{aligned}
$$

Then, the V-statistic forms of (10), (10), and (12) make clear that each has a Lipschitz constant also at most $2\nu L_k$, and hence $\hat{\zeta}_X$'s Lipschitz constant is at most $8\nu L_k$. The same is true for $\hat{\zeta}_Y$, so the Lipschitz constant of $\hat{\sigma}_\omega^2$ is at most $4(\rho_X + \rho_Y) \cdot 8\nu L_k \le 64\nu L_k$. This is indeed smaller than the $256\nu L_k$ shown by Lemma 19 of Liu et al. (2020). The exact same argument holds for $\sigma_\omega^2$, as it is simply an expectation rather than an empirical average of the same quantities. $\qquad\square$

