# OpenReview forum: "Maximum Mean Discrepancy with Unequal Sample Sizes via Generalized U-Statistics"
_TMLR — Accepted by TMLR_

### Review · Reviewer_ryE7 · 2026-02-03

**Summary Of Contributions:**

This paper addresses a practical limitation in kernel-based two-sample testing: existing methods for the Maximum Mean Discrepancy (MMD) often assume equal sample sizes, requiring wasteful data discarding in real-world unequal-size scenarios. The authors extend the theory of generalized U-statistics to derive the asymptotic distributions of the standard MMD estimator under arbitrary sample size ratios (including non-proportional regimes).

Strengthes:
1. The use of generalized U-statistics is elegant and provides a solid theoretical foundation. The results hold under minimal assumptions.

2. The detailed analysis of degeneracy (Theorem 2.7) is a valuable contribution, correcting informal claims in prior literature.

3. The proposed power optimization scheme (Section 3.3) is practical and can be integrated into existing kernel learning pipelines.

Weaknesses:
1. A more detailed discussion on the computational cost of the proposed power optimization would be beneficial.

2. The characterization of when degeneracy occurs (Theorem 2.7 parts iii-v) relies on specific conditions (continuity/analyticity of the kernel, overlap of supports).

**Audience:**

Yes

**Audience Explanation:**

The work is likely to have substantial impact in fields that rely on two-sample testing with imbalanced data, which should be of great interest to the machine learning community.

**Claims And Evidence:**

Yes

**Claims Explanation:**

This is a strong, theoretically sound paper that solves a well-known practical problem in kernel two-sample testing. The generalization via U-statistics is elegant, the results are significant, and the proposed method is practical.

**Requested Changes:**

1. Add a short intuitive explanation after Theorem 3.7 to make the result more accessible to non-specialists.

2.  Briefly discuss the computational overhead of estimating $\hat\sigma$ and optimizing kernel parameters compared to standard kernel selection

3. In experiments, briefly show sensitivity of power optimization to the choice of $\lambda$

4.  Add remarks explaining why a biased estimator of $\sigma$ is preferred. A validation experiment would help.

5. Briefly mention or cite alternative approaches to imbalanced two-sample testing beyond MMD

---

> ### Author Response · Authors · 2026-02-27
>
> Thank you for your work in reviewing our paper.
>
> > Add a short intuitive explanation after Theorem 3.7 to make the result more accessible to non-specialists.
>
> Done.
>
> > Computational cost
>
> We've added some discussion to the section "estimating the variance"; it is essentially the same as the approach of Liu et al.
>
>
> > In experiments, briefly show sensitivity of power optimization to the choice of $\lambda$
>
> We agree that this is a good idea to investigate, and intend to add this experiment. Unfortunately, we hit up against some time constraints during this rebuttal period – not having anything to do with the experiments themselves, which are not complicated – and have not yet been able to run that comparison. We expect that the sensitivity will be extremely similar to that of Liu et al.'s approach.
>
>
> > Add remarks explaining why a biased estimator of $\sigma$ is preferred.
>
> Done, in the subsection "estimating the variance."
>
>
> > Briefly mention or cite alternative approaches to imbalanced two-sample testing beyond MMD
>
> Good point; we've added some more discussion of this to the introduction.

---

### Review · Reviewer_nUAv · 2026-02-10

**Summary Of Contributions:**

The paper studies the usual unbiased estimator $\widehat{\mathrm{MMD}}^2$ when $n_X\neq n_Y$, interpreting it as a two-sample generalized U-statistic. It derives (i) a finite-sample variance expression (Thm. 3.4), (ii) asymptotic null limits beyond proportional regimes by scaling with $n:=\min(n_X,n_Y)$ (Thm. 3.7), and (iii) asymptotic normality under a non-degenerate alternative (Thm. 3.8 / Cor. 3.9). It also revisits degeneracy: infinite degeneracy occurs iff $C_P=C_Q=0$, and first-order degeneracy can occur even when $\mathrm{MMD}(P,Q)>0$; the authors provide an explicit construction and give conditions (e.g., overlapping supports with continuous kernels, or real-analytic kernels) under which degeneracy is equivalent to $\mu_P=\mu_Q$.

**Strengths**

  * Strong practical motivation, since unequal sample sizes are common, and discarding data to “balance” samples can reduce power.

* The variance decomposition for $\widehat{\mathrm{MMD}}^2$ (Thm. 3.4) is a valuable reference and clarifies how imbalance affects variance.

* The empirical checks (Figs. 1–2) convincingly support $\min(n_X,n_Y)$ as the correct scaling outside proportional asymptotics.

**Weakness**

  * I have the feeling that positioning/novelty may be overstated. Two-sample/generalized U-statistics and their (possibly unbalanced) asymptotics are classical (see e.g. https://doi.org/10.1007/s11222-021-10069-9, https://doi.org/10.1214/21-EJS1907, https://doi.org/10.48550/arXiv.1905.10651 or https://proceedings.mlr.press/v195/huang23a/huang23a.pdf). Several results feel like careful specialization/cleanup for MMD rather than fundamentally new theory. The paper should delineate more sharply what is genuinely new vs. direct consequences of existing results, and why prior work does not already cover the desired regimes.

* The paper explicitly does not handle $\mu_P\neq\mu_Q$ while the statistic is still first-order degenerate (acknowledged in Thm. 2.7(ii) and Sec. 3.2). Since kernel selection is motivated via asymptotic normality / an SNR objective, excluding this case weakens the main “power optimization” message. Especially for bounded kernels and for deep kernels (continuous but non-analytic), where disjoint-support degeneracy may be plausible (the paper itself flags this as an open point in the conclusion).

* Apparent typo: in Cor. 3.9, $\zeta_Y$ is written with $C_P$ again; shouldn't it be $\zeta_Y=\langle\mu_P-\mu_Q, C_Q(\mu_P-\mu_Q)\rangle$. Or maybe I missed something here ?

* Variance estimation for kernel learning is heuristic. The proposed plug-in $\hat\sigma$ and ridge $\lambda$ are not analyzed (bias/consistency/stability), yet this is the main actionable ingredient since thresholds are set by permutation tests anyway. Some theoretical justification or at least sensitivity analysis would clearly strengthen the contribution.

* Experiments are limited and lack key baselines. The synthetic and CIFAR experiments mainly show that “more data helps,” but do not compare against the natural baseline (choosing kernels on equal-sized subsamples, then testing with unequal sizes), nor do they report uncertainty (CIs), runtime, or enough details to reproduce the deep-kernel setup.

* Notation could be clearer, especially w.r.t. sample sizes: the generalized U-statistic depends on $(n_X,n_Y)$, while later $n$ is reused for $\min(n_X,n_Y)$. Making this distinction explicit throughout would reduce confusion.

**Additional Comments:**

None

**Audience:**

Yes

**Audience Explanation:**

Unequal sample sizes are very common in ML (domain shift, missing labels, data collection constraints), and MMD-based two-sample testing is widely used. A careful treatment of what changes when $n_X\neq n_Y$, especially variance behavior and correct asymptotic scaling, should be useful to both theory-oriented and applied readers.

The paper is also relevant to people doing kernel or representation learning for hypothesis testing, since it discusses how imbalance interacts with power, variance, and degeneracy. Even if some theory is a specialization of existing generalized U-statistic results, having the MMD-focused statements in one place is still valuable to this audience.

**Broader Impact Concerns:**

I do not see major ethical concerns. The work is methodological (two-sample testing and kernel selection) and does not directly introduce deployment risks. Typical concerns would be indirect: misuse of statistical tests to justify high-stakes decisions without proper assumptions, or fairness-related applications where distribution shift testing could be part of a pipeline.

If a broader impact statement is needed, it could briefly note that MMD-based tests can be used in sensitive settings (e.g., monitoring distribution shift across demographic groups), and emphasize that statistical significance does not imply causal explanations or fairness guarantees; users should validate assumptions and consider domain context.

**Claims And Evidence:**

Yes

**Claims Explanation:**

Mostly yes for the core theoretical claims. The paper states clear theorems (finite-sample variance, null limits outside proportional regimes using the scaling $n=\min⁡(n_X,n_Y)$, and asymptotic normality under a non-degenerate alternative) and provides proofs. The empirical figures also support the key “$n=\min⁡(n_X,n_Y)$ scaling” message in a convincing way.

That being said, some claims around “practical kernel learning” feel less solid: the variance plug-in/ridge procedure used for kernel selection is not analyzed (bias/consistency/stability), and a tricky but relevant regime is explicitly left open (first-order degeneracy even when $\mu_P\neq \mu_Q$. These gaps weaken how far the practical recommendations are supported.

**Requested Changes:**

**Critical (for acceptance, in my view):**

* Sharpen the positioning/novelty: clearly separate what is new (MMD-specific variance decomposition? non-proportional asymptotics phrased for MMD? degeneracy characterizations/constructive example?) from what follows directly from existing generalized/two-sample $U$-statistic theory; explicitly explain why prior results do not already cover the regimes targeted.

* Fix/clarify the apparent typo in Cor. 3.9 (definition of $\zeta_Y$ using $C_Q$ vs. $C_P$), or explain why the current form is correct.

* Improve experimental baselines: include the natural baseline “choose kernel on equal-sized subsamples, then test using all unequal data,” and add uncertainty reporting (e.g., confidence intervals across repetitions) plus enough implementation details for the deep-kernel setup to be reproducible.

**Strengthening (not strictly requested, but would improve the paper):**

* Provide some justification/sensitivity analysis for the variance estimator used in kernel learning (choice of plug-in, ridge $\lambda$, stability), since this is the actionable part of the method when thresholds come from permutations anyway.

* Expand discussion (and ideally partial results) on the excluded regime where $\mu_P\neq \mu_Q$ but the statistic is first-order degenerate; at minimum, give guidance on when this can happen in practice (bounded/deep kernels, disjoint supports) and how it affects kernel selection.

* Clean up notation throughout (keep $(n_X,n_Y)$) explicit; avoid reusing $n$ unless clearly redefined) and add a short “regime map” summarizing which theorem applies under which scaling/assumptions.

---

> ### Author Response · Authors · 2026-02-27
>
> Thank you in particular for your very thoughtful review, which has led to substantial improvements in the paper.
>
> > Sharpen the positioning
>
> As mentioned at the beginning of section 3, much of our work on generalized U-statistics fills in the results of Serfling (1980) section 5.5.1. The major influence of Serfling's text is prefaced in our asymptotic results in theorems 3.7/3.8 and their proofs in appendix E. One major distinction is in centering a definition of degeneracy with the following desirable properties:
> - We give simple calculations for big-$O$/$\Theta$ bounds for the variance of $U_n$, thus leading to appropriate scaling of $U_n$ in terms of $\min\{n_1,\dots,n_c\}$, which we have also empirically shown to be the "correct" alternative to $n_1+\dots+n_c$ scaling.
> - We provide a framework to find the asymptotic distribution of an appropriately scaled $U_n$ using projection-based methods.
> - Our results hold for any $c \ge 1$.
>
> It is unclear whether this version of degeneracy was previously known. Serfling's exercise 5.3.9 defines what we call the first order projection of a generalized U-statistic (with no higher order extensions), and other exercises "extend" results for U-statistics (5.2.6, 5.2.8, 5.3.12). With no further instructions nor solutions, it is ambiguous what exactly was meant, but it could likely overlap in part with our results. Our manuscript has been reworded to sharpen citations to Serfling's text.
>
> Other treatments of generalized U-statistics (Lee, van der Vaart) only consider their asymptotics in the case $c = 2$ with $n_1+\dots+n_c$ scaling, and do not discuss degeneracy or projections. For the papers you mentioned, while all are relevant (and we've added some discussion), none cover our results:
>
> - Kuntz et al. (2022): Their product form estimator, as seen in section 2.1 equation (4), is exactly a generalized U-statistic _with equal sample sizes_. Our results, which accommodate unequal sample sizes, are more general. For instance, Theorem 1 in Kuntz et al. is a specialization of our Theorem 3.8.
> - Clémençon et al. (2021): This paper makes use of maximal inequalities for stochastic processes constructed from one/two-sample U-statistics. This is not applicable to our paper, since it does not describe our null or alternative distributions.
> - Peng et al. (2021): In this text, a generalized U-statistic refers to one-sample U-statistic constructed from a random subset of the available samples. While this extends the notion of a one-sample U-statistic, it is generally not a generalized/multi-sample U-statistic as defined in section 3.1 and other classical texts.
> - Huang et al. (2023): This paper addresses the asymptotic distribution of a second-order U-statistic with data of growing dimension, while our work is in the standard fixed-dimension regime. Kernels of two arguments are relevant in our proof of theorem E.5, but our fundamental obstacle was dealing with arguments coming from two distinct distributions.
>
> Upon a more thorough search, we did also identify a previous exploration of projections of generalized U-statistics have been explored by https://link.springer.com/article/10.1007/BF01193750 (for generalized U-statistics of a very particular form related to graph theory). We also found that the treatment of asymmetric kernels in the degenerate case has been explored in https://link.springer.com/book/10.1007/978-94-017-3515-5. A discussion of existing decompositions has been added to our manuscript, and the discussion around novelty has been refined to account for this.
>
> >  typo in Cor. 3.9; clean up notation
>
> Fixed, thanks!
>
> > Improve experimental baselines
>
> We agree that the experiments could be more thorough, and plan to run more comparisons. Unfortunately, we hit up against some time constraints during this rebuttal period – not having anything to do with the experiments themselves, which are not complicated – and have not yet been able to run that comparison. We would like to emphasize that while we expect this method to perform somewhat better than pretending that the sample sizes will be equal (essentially using $\rho_X = \rho_Y = 1$), even if it ends up performing the same in practice, we think this work remains a useful theoretical contribution to the area and justifies an easy algorithmic tweak.
>
> > Provide some justification in kernel learning
>
> This is a good suggestion. As we show in a new Appendix F (and discuss at the end of section 3), the analysis of Liu et al. generalizes in a straightforward manner to our estimators; in fact, all of their theorems directly apply with even the same constants.
>
> > Expand discussion (and ideally partial results) on the excluded regime
>
> Thanks for pushing on this point as well: this regime is in fact entirely benign with respect to power optimization. We have added some discussion about this in the new subsection "Power maximization" on page 10.

---

### Review · Reviewer_sFLU · 2026-02-12

**Summary Of Contributions:**

This paper studies the two-sample testing problem with the Maximum Mean Discrepancy (MMD) when the number of samples
differs in the two populations. In particular, the authors provide an accurate characterization of the limiting distribution of
the MMD statistic, and in the meantime reveal a curious phenomenon of MMD statistics.

**Audience:**

Yes

**Audience Explanation:**

Two-sample testing problems arise ubiquitously across scientific disciplines. Developing a sharper theoretical understanding of these problems can provide principled guidance for practical applications.

**Broader Impact Concerns:**

NA.

**Claims And Evidence:**

Yes

**Claims Explanation:**

The paper provides the derivation of the limiting distribution of the MMD statistic with unequal sample sizes,
which also provides an objective for kernel selection in practice. These results support the paper's main claim.

The empirical evidence, however, is relatively weak. For example, the simulation considers only one data generating
example that is one dimensional. It would be helpful to expand on the numerical studies (see more questions in the "requested changes" section).

**Requested Changes:**

1. On page 4, it is stated "... the estimator may be first-order degenerate,
and several papers make (informal) claims to the contrary". It would be helpful
to include references that support this claim.

2. In the introduction, the authors mentioned the method by Liu et al. (2020),
and commented that ``(they) addressed this by training on equal-sized subsamples, finding
a kernel which has roughly the best test power for a test between samples of equal size, then
perhaps applying that kernel to a test set with differing sizes. While this procedure works, it is wasteful.''
I wonder whether this method is implemented in the simulation setting and compared against?


3. Related to the previous question, since the rejection threshold eventually depends on the
permutation distribution, is the only difference between the proposed method and that
proposed in Liu et al. (2020) the objective for choosing the kernel?

4. In the simulation, there seems to be no selection of kernels involved.
I am then a bit confused by what is being evaluated (it appears to the power of the permutation test as a function of $n_X$).

---

> ### Author Response · Authors · 2026-02-27
>
> Thank you for your work in reviewing our paper.
>
> > On page 4, it is stated "... the estimator may be first-order degenerate, and several papers make (informal) claims to the contrary". It would be helpful to include references that support this claim.
>
> We've added a new footnote 1 (on page 4) with a few specific pointers.
>
> > In the introduction, the authors mentioned the method by Liu et al. (2020), and commented that ``(they) addressed this by training on equal-sized subsamples, finding a kernel which has roughly the best test power for a test between samples of equal size, then perhaps applying that kernel to a test set with differing sizes. While this procedure works, it is wasteful.'' I wonder whether this method is implemented in the simulation setting and compared against?
>
> Doing the version of this that finds a test of equal size and applies to a set also of equal size is exactly the $r = 1$ case in the CIFAR experiment. We did not previously train with $r = 1$ and test with mismatched samples; we agree it would be good to do so, and plan to run this experiment.
>
> Unfortunately, we hit up against some time constraints during this rebuttal period – not having anything to do with the experiments themselves, which are not complicated – and have not yet been able to run that comparison. We would like to emphasize that while we expect this method to perform somewhat better than pretending that the sample sizes will be equal (essentially using $\rho_X = \rho_Y = 1$), even if it ends up performing the same in practice, we think this work remains a useful theoretical contribution to the area and justifies an easy algorithmic tweak.
>
>
> > Related to the previous question, since the rejection threshold eventually depends on the permutation distribution, is the only difference between the proposed method and that proposed in Liu et al. (2020) the objective for choosing the kernel?
>
> Yes, at test time, the procedure is the same; the difference applies only to the phase of selecting a kernel. Since the paper of Liu et al. is entirely about how to select a kernel, however – the foundational material on testing with a fixed kernel is much older – this is indeed a major difference. Additionally, our theoretical results are of potential interest to other problems – generalized U-statistics are a broad field – although the theory is not directly used in the procedure at test time due to the availability of permutation tests.
>
> > In the simulation, there seems to be no selection of kernels involved. I am then a bit confused by what is being evaluated (it appears to the power of the permutation test as a function of ).
>
> The synthetic results on Gaussians indeed do not involve any kernel selection; having these in the experimental section was confusing, since they serve more as motivation for the importance of using all available data. On reflection, we have removed this figure from the paper, as that portion of motivation was quite clear. The CIFAR-10.1 testing section does involve selecting a kernel via training a deep network.

---

### Decision · Action_Editor_yHeD · 2026-04-27

**Recommendation:** Accept as is

**Additional Comments:**

See the comments above.

**Audience:**

Yes

**Audience Explanation:**

The paper develops statistical theory for Maximum Mean Discrepancy (MMD) under unequal sample sizes. MMD is an important distance metric widely used in two-sample testing and other machine learning tasks, and the unequal-sample-size setting is particularly relevant in practice. These results can therefore provide useful guidance for a range of applications, including distribution comparison, model evaluation, domain adaptation, and generative model assessment. Overall, the theoretical contributions are valuable to the community and are likely to attract a solid audience at TMLR.

**Claims And Evidence:**

Yes

**Claims Explanation:**

The theoretical results in the paper are supported by rigorous proofs, and the authors also provide some experiments to validate the theory.